# Simple descriptor derived from symbolic regression accelerating the discovery of new perovskite catalysts

Baicheng Weng[1,2,3,6], Zhilong Song[2,6], Rilong Zhu[4], Qingyu Yan[4], Qingde Sun[2], Corey G. Grice[1], Yanfa Yan [1✉] & Wan-Jian Yin [2,5✉]

Symbolic regression (SR) is an approach of interpretable machine learning for building mathematical formulas that best fit certain datasets. In this work, SR is used to guide the design of new oxide perovskite catalysts with improved oxygen evolution reaction (OER) activities. A simple descriptor, $\mu/t$, where $\mu$ and $t$ are the octahedral and tolerance factors, respectively, is identified, which accelerates the discovery of a series of new oxide perovskite catalysts with improved OER activity. We successfully synthesise five new oxide perovskites and characterise their OER activities. Remarkably, four of them, $Cs_{0.4}La_{0.6}Mn_{0.25}Co_{0.75}O_3$, $Cs_{0.3}La_{0.7}NiO_3$, $SrNi_{0.75}Co_{0.25}O_3$, and $Sr_{0.25}Ba_{0.75}NiO_3$, are among the oxide perovskite catalysts with the highest intrinsic activities. Our results demonstrate the potential of SR for accelerating the data-driven design and discovery of new materials with improved properties.

[1] Department of Physics & Astronomy, and Wright Center for Photovoltaics Innovation and Commercialization, The University of Toledo, Toledo, OH 43606, USA. [2] College of Energy, Soochow Institute for Energy and Materials InnovationS (SIEMIS), and Jiangsu Provincial Key Laboratory for Advanced Carbon Materials and Wearable Energy Technologies, Soochow University, 215006 Suzhou, China. [3] College of Chemistry and Chemical Engineering, Central South University, 410083 Changsha, China. [4] College of Chemistry and Chemical Engineering, Hunan University, 410082 Changsha, China. [5] Key Lab of Advanced Optical Manufacturing Technologies of Jiangsu Province & Key Lab of Modern Optical Technologies of Education Ministry of China, Soochow University, 215006 Suzhou, China. [6] These authors contributed equally: Baicheng Weng, Zhilong Song. ✉email: yanfa.yan@utoledo.edu; wjyin@suda.edu.cn

Machine learning (ML) is increasingly used in the field of materials informatics as an effective tool for discovering quantitative structure— or composition—property relationships that can accelerate materials design[1–5]. However, the black-box model of ML is often criticized not able to provide new "physical laws", which limits its potential in certain cases[6,7]. Symbolic regression (SR) is an approach of interpretable machine learning that simultaneously searches for the optimal mathematical formula of a function and set of parameters in the function[1,8]. Therefore, SR is capable to deliver interpretable mathematical formulas that may provide direct guidance for materials design. Despite the great potential, the application in the field of material science is still limited.

In this communication, we demonstrate that SR can construct a simple descriptor that enables the acceleration of the materials discovery for oxide perovskite catalysts. Oxide perovskites (ABO$_3$) are an important family of catalysts for OER applications[9,10], which are in high demand for renewable energy production and storage, such as hydrogen production from water-splitting[11] and rechargeable metal-air batteries[12], because of their structural flexibility, compositional versatility, and chemical stability[13]. Moreover, oxide perovskites have recently been extended to the bifunctional application of OER and oxygen reduction reaction[14,15]. The catalysis activities of oxide perovskite catalysts can be described by descriptors, as demonstrated by various studies over the past sixty years. Several descriptors, such as the reaction free energy[16,17] and $e_g$ occupancy[9,18], have been successfully used to understand the trend of OER activity and achieved great success in this regard. Nevertheless, those descriptors require prior knowledge based on density functional theory (DFT) calculations therefore bear limited applicability to design new materials, where DFT-calculated values are unknown a priori and highly dependent on the used methodologies[19]. Meanwhile, it is difficult for DFT calculation to accurately determine $e_g$ occupancy where the surface spin state is not well known[20]. A good descriptor should be simple and yet provide physical insight[21], which will guide and accelerate the discovery of new perovskite oxide OER catalysts. In this work, we propose that SR is perfectly suitable for identifying suitable descriptor to accelerating the discovery of new perovskite catalysts.

Figure 1 shows the workflow diagram of this study. SR analysis may not require massive datasets, if the datasets used are consistent and reliable[1,22]. Therefore, we firstly synthesise 18 well-studied oxide perovskite catalysts to produce consistent and comparable datasets of OER activity for SR analysis. A descriptor with the balance of simplicity and accuracy is then chosen and help develop strategies to accelerate the discovery of new oxide perovskites. The generality of the descriptor is confirmed by analysing data reported independently by other research groups. Based on this descriptor, materials screening is conducted to search for new oxide perovskite catalysts with improved OER activities. To validate the predictions, a few numbers of new oxide perovskites with potentially high OER activity are synthesised and their OER activities are characterised and compared with their predicted values and those of current state-of-the-art oxide perovskite catalysts.

## Results

**Data acquisition.** Comparable training data used in SR analysis are of crucial importance for SR in order to produce useful mathematical formulas[23]. Since the first discovery of oxide perovskite LaNiO$_3$ as OER catalyst in 1970s[24], the chemical management of A- or B-site cations has been used to tune the OER activity, permitted by the structural and chemical flexibility of perovskite structures. The results reported by different groups and produced under different experimental conditions over a period of half a century are summarised in a recent review article[13]. However, the comparability of those data is doubtful due to different environments of experiments and measurements. To ensure meaningful and valuable SR analysis, we synthesised eighteen known oxide perovskite catalysts (Supplementary Fig. 1; Supplementary Table 1). Four samples were made for each perovskite and OER measurement was conducted three times under the same conditions with freshly made catalyst inks. Four each measurement, the $V_{RHE}$ values at five current densities of 50 μA cm$^{-2}$, 5 mA cm$^{-2}$, 10 mA cm$^{-2}$, 15 mA cm$^{-2}$ and 20 mA cm$^{-2}$ in linear sweep voltammetry (LSV) curve were adopted for SR analysis. Therefore, there are totally 18 perovskites × 4 samples × 3 measurements × 5 current densities = 1080 data points (Fig. 2a). The values were then normalised by the catalyst loading concentration and Brunauer–Emmet–Teller (BET) surface area (Supplementary Table 2) and shown in Fig. 2a and Supplementary Data 1–5. Details of the materials synthesis, along with the structural and OER characterisation, can be found in the

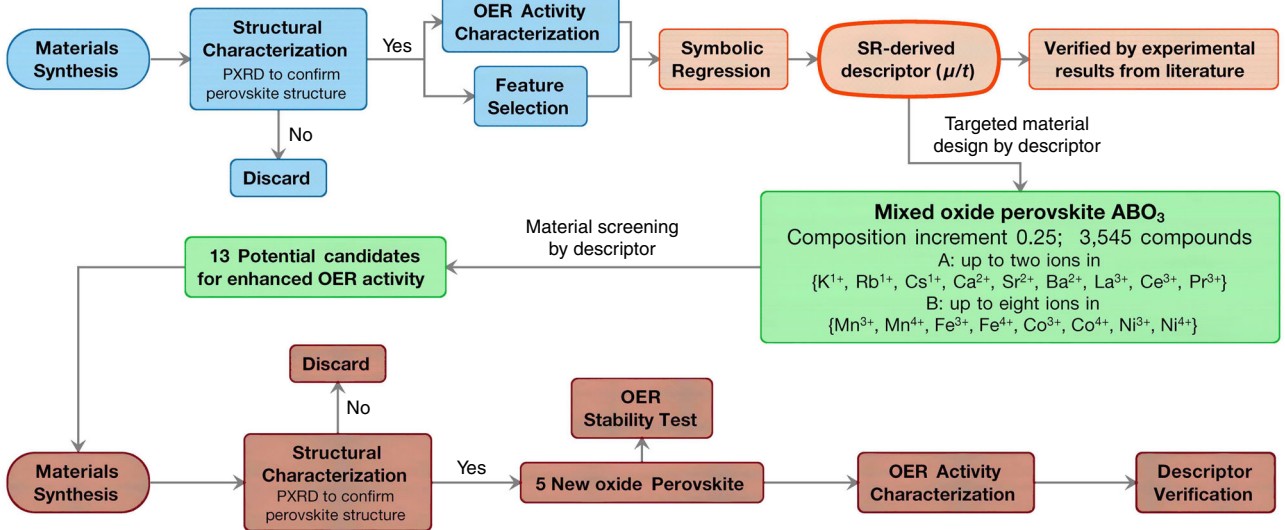

**Fig. 1 Workflow diagram.** It contains four major parts: dataset generation (blue), SR (red), materials design and screening (green) and experimental verification (brown).

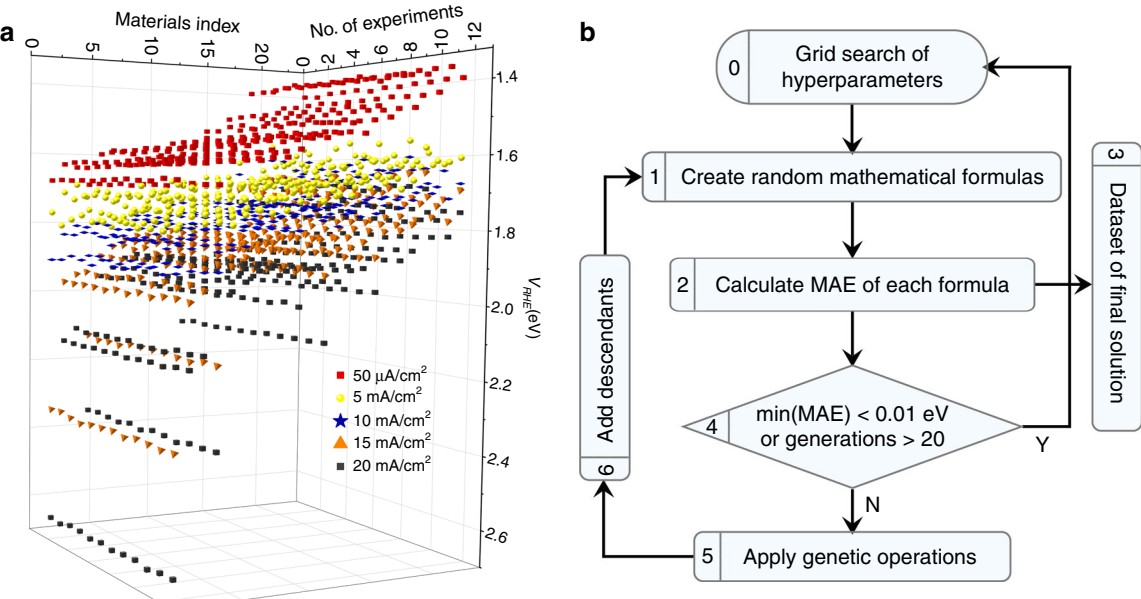

**Fig. 2 Data collection and process. a** The landscape of all $V_{RHE}$ data produced by experiments, including eighteen conventional and five new perovskites (totally twenty-three perovskites listed as 'Materials index' with sequence shown in Table 1). Each perovskite has been made four samples and each sample has been measured three times (totally twelve measurements listed as 'No. of measurements'). For each measurement, we adopted $V_{RHE}$ values at five current densities of 50 μA cm$^{-2}$, 5 mA cm$^{-2}$, 10 mA cm$^{-2}$, 15 mA cm$^{-2}$, and 20 mA cm$^{-2}$. The exact values of those data points are provided in Supplementary Data 1–5. **b** The flowchart of symbolic regression based on genetic programming (see more details of this flowchart and SR in Supplementary Information).

"Methods". Seven of these oxide perovskite catalysts were also reported by Suntvich et al.[9] and the results from both groups showed the same trend in $V_{RHE}$ values (Supplementary Fig. 2), although the absolute values are slightly different.

**SR training**. With the available experimental data shown in Fig. 2a, SR was then adopted to construct mathematical formula linking the materials parameters and $V_{RHE}$. To ensure that the SR analysis determines mathematical formulas that are useful for our purpose, it is critical to select relevant parameters to be included in the mathematical formulas based on prior knowledge[1]. Considering the importance of previous descriptors[3,9,12,13,25,26], we chose electronic parameters such as the number of $d$ electrons for TM ions ($N_d$), electronegativity values $\chi_A$ and $\chi_B$, and valence states $Q_A$, as well as structural parameters such as ionic radii $R_A$, the tolerance factor $t$, and the octahedral factor $\mu$, where A and B refer to the A- and B-site cations, respectively (Table 1). The tolerance factor $t$, defined as $\frac{r_A+r_O}{\sqrt{2}(r_B+r_O)}$ and octahedral factor $\mu$, defined as $r_B/r_O$, are commonly used features in ML studies of perovskites[23,27,28].

The mathematical formulas were then generated and selected by using SR with genetic programming (GPSR) as implemented in gplearn code[29]. The flowchart of GPSR process in this work is described in Fig. 2b. In this work, SR initially builds a population of random mathematical formulas with these parameters as variables. Then, these mathematical formulas breed, mutate, and evolve to form new ones via genetic programming. The derived mathematical formulas compete to model experimental data by evaluating the mean absolute errors (MAEs) between the predicted and experimental $V_{RHE}$. A grid search of hyperparameters resulted in ~8640 mathematical formulas (descriptors), which were characterised by their MAE's and complexities, as described in Fig. 3a. The hyperparameters setup can be found in Method part and extended information about GPSR can be found in Supporting Information.

**Descriptor generation and analysis**. Of the produced descriptors, only those with low MAE (high accuracy) and low complexity are suitable for guiding the discovery of new oxide perovskite catalysts. The nine mathematical formulas at the Pareto front [marked as A–I in Fig. 3a] that met the criteria of simplicity and accuracy among the 43,200,000 candidates are shown in Table 2. Among them, $\mu/t$ is the best compromise between complexity and accuracy. To clearly show the correlation, the $V_{RHE}$ at current densities of 5 mA cm$^{-2}$ are shown in terms of $\mu/t$ in Fig. 3b. For each perovskite, the average values and error bars are the experimental uncertainties from 12 measurement data (4 samples with each 3 measurements). Interestingly, it shows a linear and monotonic behaviour instead of prevalent volcano shape for conventional descriptors. Such linear correlations remain at other current densities, i.e. 50 μA cm$^{-2}$, 10 mA cm$^{-2}$, 15 mA cm$^{-2}$ and 20 mA cm$^{-2}$ as shown in Supplementary Fig. 3. To further verify the generality of this descriptor, we used $\mu/t$ to fit the experimental work[9] originally reporting the volcano shape for descriptor $e_g$ (Fig. 3c). As shown in Fig. 3d, $\mu/t$ provided a clear linear and monotonic correlation with $V_{RHE}$, with MAE comparable to the volcano shape for descriptor $e_g$. Apart from the seminal work of ref. [9], the generality of $\mu/t$ can be also confirmed by recent works[30–32] as their data reorganized in Supplementary Fig. 4. For experimental data spanning over sixty years from different groups (Table 6 of ref. [13]), their $V_{RHE}$ values are reorganized according to their $\mu/t$ values; despite some discrepancies, a roughly linear correlation was observed for the majority of the data points (Supplementary Fig. 5). Such good correlation reveals that the SR-derived descriptors, e.g., $\mu/t$ indeed provide meaningful insights for OER activity of oxide perovskites.

The descriptor $\mu/t$ reveals that the OER activity of oxide perovskite catalysts is closely related to the structural factors of the catalysts; i.e. a smaller $\mu$ and a larger $t$ should lead to higher OER activity. Such a simple descriptor is superior to conventional descriptors since it does not require additional DFT calculations and can be directly used for materials design. Accordingly, we

**Table 1 Key materials parameters of 23 selected oxide perovskites.**

| No. | Materials | $t$ | $\mu$ | $R_A$(Å) | $\chi_A$ | $\chi_B$ | $Q_A$ | $N_d$ | $\mu/t$ |
|---|---|---|---|---|---|---|---|---|---|
| *Conventional Perovskites* | | | | | | | | | |
| 1 | $LaMnO_3$ | 0.993 | 0.430 | 1.36 | 1.1 | 1.55 | 3 | 4 | 0.433 |
| 2 | $LaMn_{0.5}Ni_{0.5}O_3$ | 0.998 | 0.422 | 1.36 | 1.1 | 1.73 | 3 | 5.5 | 0.423 |
| 3 | $LaNiO_3$ | 1.003 | 0.415 | 1.36 | 1.1 | 1.91 | 3 | 7 | 0.413 |
| 4 | $LaMn_{0.5}Cu_{0.5}O_3$ | 0.988 | 0.437 | 1.36 | 1.1 | 1.725 | 3 | 6 | 0.442 |
| 5 | $LaNi_{0.9}Fe_{0.1}O_3$ | 1.004 | 0.414 | 1.36 | 1.1 | 1.902 | 3 | 6.8 | 0.413 |
| 6 | $LaNi_{0.8}Fe_{0.2}O_3$ | 1.004 | 0.413 | 1.36 | 1.1 | 1.894 | 3 | 6.6 | 0.412 |
| 7 | $LaFeO_3$ | 1.009 | 0.407 | 1.36 | 1.1 | 1.83 | 3 | 5 | 0.404 |
| 8 | $La_{0.5}Pr_{0.5}FeO_3$ | 1.010 | 0.407 | 1.365 | 1.115 | 1.83 | 3 | 5 | 0.403 |
| 9 | $PrFeO_3$ | 1.012 | 0.407 | 1.37 | 1.13 | 1.83 | 3 | 5 | 0.402 |
| 10 | $LaCoO_3$ | 1.011 | 0.404 | 1.36 | 1.1 | 1.88 | 3 | 6 | 0.399 |
| 11 | $La_{0.5}Ca_{0.5}CoO_3$ | 1.011 | 0.398 | 1.35 | 1.05 | 1.88 | 2.5 | 5.5 | 0.394 |
| 12 | $La_{0.8}Sr_{0.2}CoO_3$ | 1.019 | 0.401 | 1.376 | 1.07 | 1.88 | 2.8 | 5.8 | 0.394 |
| 13 | $Sr_{0.25}La_{0.75}Fe_{0.5}Co_{0.5}O_3$ | 1.020 | 0.401 | 1.38 | 1.063 | 1.855 | 2.75 | 5.25 | 0.393 |
| 14 | $La_{0.4}Sr_{0.6}CoO_3$ | 1.034 | 0.397 | 1.408 | 1.01 | 1.88 | 2.4 | 5.4 | 0.384 |
| 15 | $La_{0.2}Sr_{0.8}CoO_3$ | 1.042 | 0.395 | 1.424 | 0.98 | 1.88 | 2.2 | 5.2 | 0.379 |
| 16 | $SrCoO_3$ | 1.049 | 0.393 | 1.44 | 0.95 | 1.88 | 2 | 5 | 0.374 |
| 17 | $Ba_{0.5}Sr_{0.5}Co_{0.8}Fe_{0.2}O_3$ | 1.082 | 0.391 | 1.525 | 0.92 | 1.876 | 2 | 4.8 | 0.361 |
| 18 | $BaFeO_3$ | 1.119 | 0.385 | 1.61 | 0.89 | 1.83 | 2 | 4 | 0.344 |
| *New Perovskites* | | | | | | | | | |
| 19 | $Cs_{0.25}La_{0.75}Mn_{0.5}Ni_{0.5}O_3$ | 1.064 | 0.398 | 1.49 | 1.023 | 1.73 | 2.5 | 5 | 0.374 |
| 20 | $Cs_{0.4}La_{0.6}Mn_{0.25}Co_{0.75}O_3$ | 1.095 | 0.395 | 1.568 | 0.976 | 1.798 | 2.2 | 4.7 | 0.361 |
| 21 | $Cs_{0.3}La_{0.7}NiO_3$ | 1.088 | 0.379 | 1.516 | 1.007 | 1.91 | 2.4 | 6.4 | 0.348 |
| 22 | $SrNi_{0.75}Co_{0.25}O_3$ | 1.071 | 0.365 | 1.44 | 0.95 | 1.903 | 2 | 5.75 | 0.341 |
| 23 | $Sr_{0.25}Ba_{0.75}NiO_3$ | 1.127 | 0.356 | 1.568 | 0.905 | 1.91 | 2 | 6 | 0.315 |

The key materials parameters include the tolerance factor ($t$), octahedral factor ($\mu$), ionic radii of A-site ($R_A$) and B-site ($R_B$), electronegativity of A-site ($\chi_A$) and B-site ($\chi_B$), valence state of A-site ($Q_A$), and number of $d$ electrons on TM B-site ($N_d$). The materials are ordered by the value of $\mu/t$ in each dataset of conventional and new perovskites.(See Supplementary Table 7 for calculation details.).

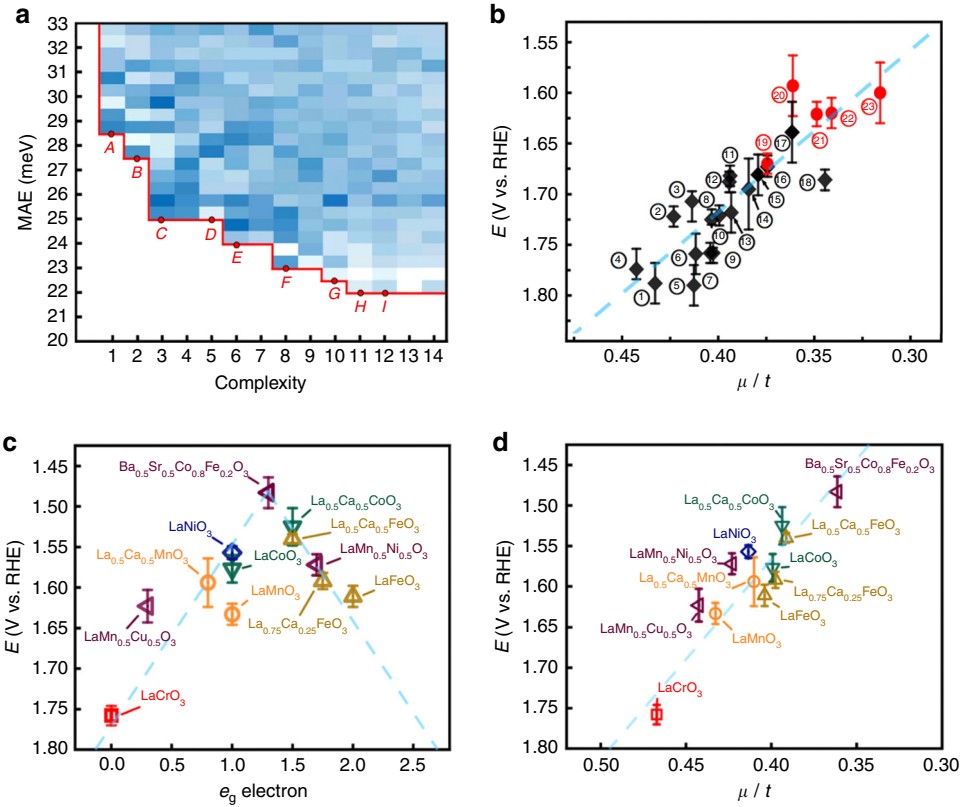

**Fig. 3 Descriptor generation and performance. a** Pareto front of MAE vs. complexity of 8640 mathematical formulas shown via density plot. **b** $V_{RHE}$ vs. $\mu/t$ (black diamonds: conventional perovskites; red dots: new perovskites). The current densities are normalised by BET surface areas (Supplementary Table 2) and loading amount. **c** Figure 2 from the study of Suntivich et al.[9] reproduced with permission from the American Association for the Advancement of Science. **d** Reformatted plot according to descriptor $\mu/t$ of **c**. The MAE (Pearson correlation coefficient) for **c**, **d** were 20.6 meV (0.923) and 21.0 meV (0.928), respectively.(The error bar in **b** is produced by the maximum and minimum values in experiments data).

**Table 2 The nine mathematical formulas at the Pareto front in Fig. 3a.**

| Point | Formulas | MAE (eV) | Complexity |
|-------|----------|----------|------------|
| A | $\frac{1.751}{t}$ | 0.0286 | 1 |
| B | $\left(\frac{1.992\mu}{0.276}\right)^{0.5}$ | 0.0279 | 2 |
| C | $1.554\frac{\mu}{t} + 1.092$ | 0.0253 | 3 |
| D | $\left(t + 0.289 + Q_A^{0.5}\right)^{0.5}$ | 0.0252 | 5 |
| E | $\left(1.282 + Q_A^{0.5}\right)^{0.5} + \mu$ | 0.0244 | 6 |
| F | $\left(\frac{Q_A^{0.5}}{\chi_B} + 1.034 + \chi_A^{0.5}\right)^{0.5}$ | 0.0232 | 8 |
| G | $\left(\frac{\frac{Q_A^{0.5}}{t}}{\chi_B} + 1.034 + \chi_A^{0.5}\right)^{0.5}$ | 0.0225 | 10 |
| H | $\left(\frac{Q_A^{0.5}}{\chi_B} + 1.034 + (Q_A\mu)^{0.5}\right)^{0.5}$ | 0.0224 | 11 |
| I | $\left(\frac{Q_A^{0.5}}{\chi_B} + 1 + 0.34\mu + \left(\frac{\mu}{t}\right)^{0.5}\right)^{0.5}$ | 0.0220 | 12 |

used a rational strategy to accelerate the screening process: adopting large cations on the A site (increasing $t$) and small cations on the B site (decreasing $\mu$). Previously, the commonly used A-site cations in oxide perovskite catalysts are group IIA (Ca, Sr, Ba) and group IIIB (La, Ce, Pr) elements[13]. Based on the insight of the new descriptor developed here, we considered incorporating large group-IA elements (K, Rb, Cs) onto the A site to increase $t$. Among the TM ions that can form perovskite oxides, $3d$ TM ions have the smallest ionic radii, which is consistent with the fact that all existing active oxide perovskite catalysts contain Mn, Fe, Co, and Ni cations (the smallest among the $3d$ TM ions) on the B site. $4d/5d$ TM oxide perovskites are catalytically less active, despite having similar $d$ electron configurations. Therefore, we considered that the A site contains up to two ions from (K$^{1+}$, Rb$^{1+}$, Cs$^{1+}$, Ca$^{2+}$, Sr$^{2+}$, Ba$^{2+}$, La$^{3+}$, Ce$^{3+}$, Pr$^{3+}$) and the B site contains up to eight ions from (Mn$^{3+}$, Mn$^{4+}$, Fe$^{3+}$, Fe$^{4+}$, Co$^{3+}$, Co$^{4+}$, Ni$^{3+}$, Ni$^{4+}$) with variation in an increment of 0.25 for the A and B ionic ratio. Note that the actual stoichiometric ratios depend on the synthesis conditions and the formability of the target perovskites. Subject to the requirement of charge balance, 3,545 oxide perovskites were obtained and their $\mu/t$ values were calculated. These oxide perovskites are listed in Supplementary Data 6 in order of increasing $\mu/t$ value. There are many new oxide perovskites with $\mu/t$ values smaller than those of materials reported in the literature, revealing a new and large group of previously unexplored OER catalysts.

**Screening, synthesis and characterisation of new oxide perovskite catalysts.** The formability and stabilities of 3545 oxide perovskites have not been verified. Therefore, we selected thirteen new oxide perovskites in the smallest $\mu/t$ values (the topmost region in Supplementary Data 6) with an increment of ~0.015 in $\mu/t$ values to consider sufficient elemental and compositional diversity for experimental verification. These thirteen perovskite oxides are: Ba$_{0.75}$Sr$_{0.25}$NiO$_3$, Cs$_{0.4}$La$_{0.6}$Mn$_{0.25}$Co$_{0.75}$O$_3$, SrNi$_{0.75}$Co$_{0.25}$O$_3$, Cs$_{0.3}$La$_{0.7}$NiO$_3$, Cs$_{0.25}$La$_{0.75}$Mn$_{0.5}$Ni$_{0.5}$O$_3$, Cs$_{0.5}$La$_{0.5}$Mn$_{0.5}$Ni$_{0.5}$O$_3$, Sr$_{0.25}$La$_{0.75}$Mn$_{0.5}$Fe$_{0.5}$O$_3$, Ba$_{0.75}$Pr$_{0.25}$Ni$_{0.5}$Fe$_{0.5}$O$_3$, Cs$_{0.6}$La$_{0.4}$Mn$_{0.75}$Co$_{0.25}$O$_3$, Cs$_{0.5}$La$_{0.5}$MnO$_3$, Cs$_{0.5}$La$_{0.5}$Mn$_{0.25}$Co$_{0.75}$O$_3$, Cs$_{0.5}$La$_{0.5}$Mn$_{0.5}$Co$_{0.5}$O$_3$, and Cs$_{0.25}$Pr$_{0.75}$Mn$_{0.25}$Fe$_{0.25}$Co$_{0.25}$Ni$_{0.25}$O$_3$. The synthesis method is described in detail in the Methods section. We found that eight of them contained significant amounts of impurity or secondary phases, as indicated by the asterisks in the powder X-ray diffraction (PXRD) patterns (Supplementary Fig. 6). For example, Cs$_{0.5}$La$_{0.5}$Mn$_{0.5}$Ni$_{0.5}$O$_3$, Cs$_{0.6}$La$_{0.4}$Mn$_{0.75}$Co$_{0.25}$O$_3$,

Cs$_{0.5}$La$_{0.5}$MnO$_3$, Cs$_{0.5}$La$_{0.5}$Mn$_{0.25}$Co$_{0.75}$O$_3$, and Cs$_{0.5}$La$_{0.5}$Mn$_{0.5}$Co$_{0.5}$O$_3$ showed an impurity phase of MnO$_{4+\delta}$ (main diffraction peaks at 12° and 24°). Ba$_{0.75}$Pr$_{0.25}$Ni$_{0.5}$Fe$_{0.5}$O$_3$ contained Pr$_2$O$_3$ and NiO impurity phases. Five compounds including Cs$_{0.4}$La$_{0.6}$Mn$_{0.25}$Co$_{0.75}$O$_3$, Cs$_{0.3}$La$_{0.7}$NiO$_3$, Cs$_{0.25}$La$_{0.75}$Mn$_{0.5}$Ni$_{0.5}$O$_3$, Sr$_{0.25}$Ba$_{0.75}$NiO$_3$, and SrNi$_{0.75}$Co$_{0.25}$O$_3$, formed pure perovskite phases, as by confirmed PXRD (Supplementary Fig. 6). The OER activities of these five new pure oxide perovskites were then characterised (Fig. 4a–c). Cs$_{0.4}$La$_{0.6}$Mn$_{0.25}$Co$_{0.75}$O$_3$, Cs$_{0.3}$La$_{0.7}$NiO$_3$, SrNi$_{0.75}$Co$_{0.25}$O$_3$, and Sr$_{0.25}$Ba$_{0.75}$NiO$_3$ showed lower $V_{RHE}$ values (higher OER activity) than BSCF did. The specific activities are also compared with the state-of-the-art perovskite oxide catalysts[20]. We found that our materials are among the oxide perovskite catalysts with the highest specific activities[10] (Supplementary Fig. 7). Remarkably, the experimental $V_{RHE}$ values of these new oxide perovskite catalysts follow the same trend of SR-derived descriptor, $\mu/t$, as shown in Fig. 3b. To further verify the descriptor, the SR procedure is repeated with the inclusion of five new predicted perovskites. Most of derived mathematical formulas that had been residing near the Pareto front (Fig. 3a), including $\mu/t$, remain (Supplementary Fig. 8 and Supplementary Table 3); this persistence shows that the addition of more training examples does not generate a significant alteration in the model's response, indicating that the model remained predictive with these new perovskites. It is worth noting that we have selected a very limited number of compositions for experimental synthesis and characterisation because of limited resources. It is highly anticipated that more of these predicted oxide perovskite catalysts with high OER activities can be experimentally synthesised and their OER activities will be verified.

The stability of the four new oxide perovskite catalysts with OER activities higher than previously reported oxide perovskite catalysts were tested galvanostatically at 10 mA·cm$^{-2}$ disk current (Fig. 4d). We selected a higher disk current density for stability testing to verify the activity decay under strong polarisation conditions. Cs$_{0.4}$La$_{0.6}$Mn$_{0.25}$Co$_{0.75}$O$_3$, Cs$_{0.3}$La$_{0.7}$NiO$_3$, SrNi$_{0.75}$Co$_{0.25}$O$_3$, and Sr$_{0.25}$Ba$_{0.75}$NiO$_3$ showed lower activity degradation than BSCF. In particular, the Sr$_{0.25}$Ba$_{0.75}$NiO$_3$ electrode maintained a stable $V_{RHE}$ over 12 h of stability testing without significant decay. Under the same conditions, the BSCF sample showed a much faster degradation rate, with only 90% retention after 9 h. After OER durability tests, the Sr$_{0.25}$Ba$_{0.75}$NiO$_3$ electrode maintained its original morphology. Scanning transmission electron microscopy (STEM) and high-resolution transmission electron microscopy images revealed no significant surface amorphization. The surfaces of the Sr$_{0.25}$Ba$_{0.75}$NiO$_3$ particles maintained good crystallinity after stability tests, as confirmed by clear observation of the same lattice spacings (Fig. 5) and elemental analysis (Supplementary Table 4). Recent work has shown that increasing the valence states of $3d$-TMs such as Ni and Co from 2+/3+ to 3+/4+ can boost the OER activities of LaCoO$_3$ and LaNiO$_3$[33]. Interestingly, apart from increasing $t$, Cs$^{1+}$ substitution on the A site is a viable route to enhance the valence states of TM B-site ions in oxide perovskites. This correlates with the SR-derived descriptor, $\mu/t$, since increasing valence states inevitably reduces the ionic radii of TMs, which in turn reduces the $\mu$ value, and, therefore, reduces $\mu/t$. Meanwhile, recent theoretical reports predicted that SrNiO$_3$ should have high OER activity[34]. Unfortunately, the hexagonal close packing of Sr and O atoms prevents the formation of the perovskite structure. To mitigate this issue, La was proposed to partially substitute Sr. However, partial La substitution leads to the formation of a Ruddlesden–Popper crystal structure instead of perovskite structures[31]. Interestingly, the descriptor $\mu/t$ suggests that partial substitution of Sr using larger Ba atoms can enhance catalytic

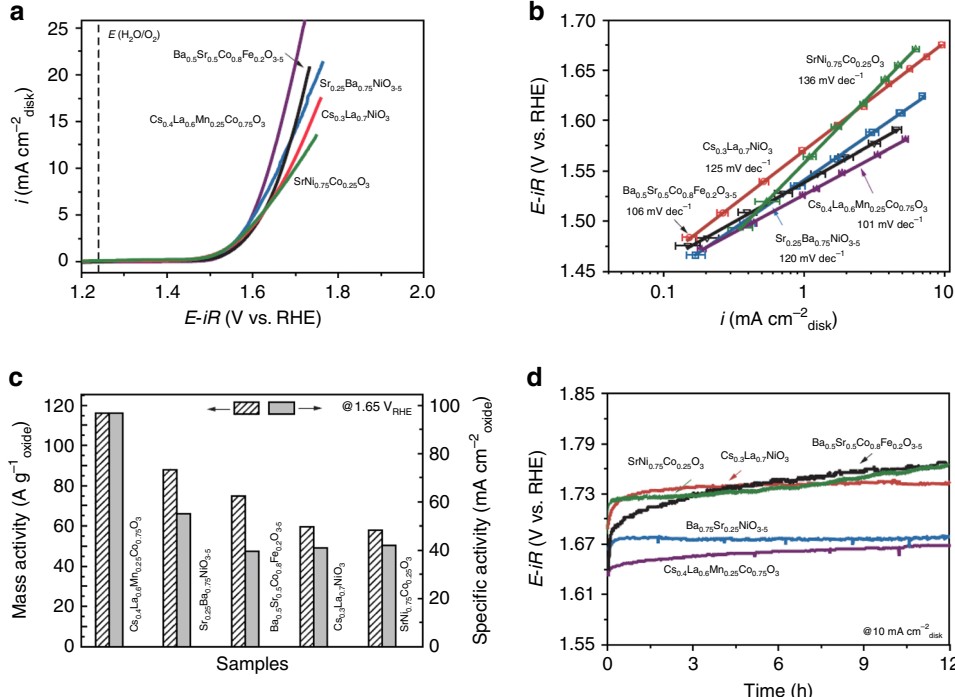

**Fig. 4 OER characterisations of Ba$_{0.5}$Sr$_{0.5}$Co$_{0.8}$Fe$_{0.2}$O$_3$ and predicted new oxide perovskites. a** LSV curves. **b** Corresponding Tafel slopes. **c** Mass and specific activities. **d** Results of stability tests under galvanostatic conditions at 10 mA cm$^{-2}$ disk current density.

activity. Our experiments showed that Ba$_{0.75}$Sr$_{0.25}$NiO$_3$ can be synthesised with the perovskite structure and its OER activity is even higher than BSCF (Fig. 5), demonstrating the usefulness of the SR-derived descriptor.

## Discussion

Those results show that even with a small dataset, the SR analysis could provide simple and meaningful descriptors that enabled us to discover new oxide OER catalysts with improved activities, which is consistent with successful application of small data in materials design by adaptive ML[4,5]. The descriptor of $\mu/t$ implies that the catalytic activity of oxide perovskites is closely related to their structural stability, i.e. a lower stability leads to a high activity. Feature analysis in SR process shows that $\mu$, $t$, and $Q_A$ correlate with the catalytic activity more than $R_A$, $N_d$, $\chi_A$, and $\chi_B$ (Supplementary Fig. 9). Considering the $t$ and $\mu$ are functions of $r_A$ and $r_B$, we also trained SR model based on the parameters of $r_A$, $r_B$, $N_d$, $\chi_A$, $\chi_B$, $Q_A$ without $t$, $\mu$. The results are shown in Supplementary Fig. 13 and Supplementary Table 5. However, the MAE of descriptors at the same complexity on Pareto front are mostly larger than the descriptors discovered based on $\mu$, $t$, $r_A$, $N_d$, $\chi_A$, $\chi_B$, $Q_A$. The oxide perovskites showing improved OER activity had $t > 1$ (Table 1 and also Table 6 in ref. [13]), which were considered unstable perovskites[35]. However, we found that these perovskites could be synthesised under suitable conditions. Notably,, we exhaustively searched Inorganic Crystal Structure Database(ICSD) and found that the existing oxide perovskites mostly have $t < 0.95$ and $\mu > 0.55$ (Supplementary Fig. 14). However, oxide perovskites reported to be catalyst in the last forty years lie in a small confined range ($t > 0.95$ and $\mu < 0.55$). According to the descriptor of $\mu/t$, most of oxide perovskites are less catalytically active, which seems consistent with existing experimental results that oxide perovskite catalysts are limited in a few types of perovskites[10]. More in-depth understanding of

correlation among $\mu/t$, catalysis activity and structural stability is out of scope of current research but deserves further study.

In summary, we used SR to identify a simple descriptor for describing the OER activity of oxide perovskite catalysts. This simple descriptor quantitatively predicted the OER activity of oxide perovskites and enabled us to rapidly discover a series of new oxide perovskite catalysts with improved OER activities. For proof of concept, we successfully synthesised five oxide perovskites and four of them exhibited OER activities surpassing those of existing oxide perovskite catalysts reported in the literature. We anticipate that more of the predicted new oxide perovskite catalysts can be synthesised and their OER activities verified. Our results demonstrate that SR is a powerful ML technique to discover physically meaningful descriptors when sufficient comparable data is available. This work suggests a new direction for discovering functional materials with improved activities.

## Methods

**Symbolic regression**. Symbolic regression analysis using a genetic algorithm was performed using gplearn[29], a Python library that extends scikit-learn, a machine learning tool, for symbolic regression. The hyper-parameters setup for gplearn is listed in Table 3. The explanation of each hyper-parameter in Table 3 are following:

The meanings of genetic operations of pc, ps, ph, and pp above can be found in Supplementary Fig. 10. The grid search method was used for pc, ps, and parsimony coefficient. As shown in the Table 3, there are 18 pc values from 0.5 to 0.95 with step of 0.025, 8 ps values and 3 parsimony coefficients. Therefore, a grid search contains $18 \times 8 \times 3 = 432$ hyper-parameters. More information about SR can be found in the Supplementary Information.

**Experimental synthesis of oxide perovskites**. The oxide perovskites were synthesised using a modified Pechini method following by thermal calcination at 850–1000 °C under dry air/oxygen atmospheres. Briefly, the acetate or nitrate precursors of the perovskite oxides (4 mmol) were mixed in methanol/H$_2$O (10 mL, 2:1 $v$:$v$), and citric acid (10 mmol) was added to obtain a clear sol. The mixture was dried at 120 °C and the remaining solid was calcinated at 500 °C for 1 h in air. Then, the obtained powder was ground into fine powder and pressed into pellets with a diameter of 15 mm using a hydraulic press at 20 MPa. Finally,

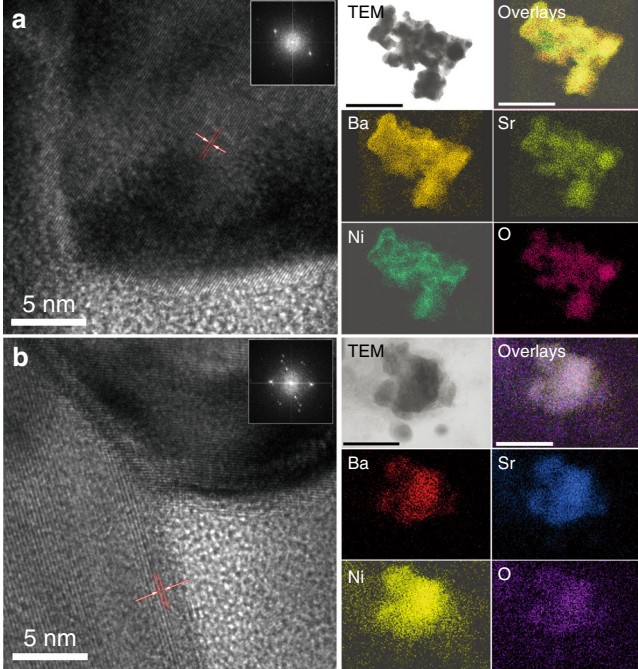

**Fig. 5 Morphology measurements of $Ba_{0.75}Sr_{0.25}NiO_3$ before and after OER testing. a** HRTEM before a stability test. **b** HRTEM after a stability test. Right side: STEM atomic mapping (scale bar: 500 nm). The labelled lattice spacing is around 0.3 nm, which corresponded to the (110) lattice planes of $Ba_{0.75}Sr_{0.25}NiO_3$, in good agreement with the PXRD measurements. The insets of **a**, **b** show the fast Fourier transform image of the corresponding HRTEM image. The well-regulated arrayed spots indicated that the grown crystal had high crystallinity. HRTEM of $Ba_{0.75}Sr_{0.25}NiO_3$ before and after OER testing clearly showed the same lattice spacing and very similar fast Fourier transform images, suggesting outstanding stability of the $Ba_{0.75}Sr_{0.25}NiO_3$ sample under OER conditions. The maintenance of good crystallinity indicates that $Ba_{0.75}Sr_{0.25}NiO_3$ is a stable OER electrocatalyst. In order to verify the atomic distribution, STEM mapping was conducted; the even distribution of the atoms over the analyzed area further demonstrates the excellent stability of the sample.

the pellets were calcinated at 850–1000 °C for 6 h under dry air/oxygen atmospheres.

**Crystal structure characterisation.** The structure and phase of the synthesised materials were examined by PXRD (Ultima III, Rigaku, Japan) and Raman spectroscopy (Bruker FT Raman Spectrometer with a laser wavelength of 532 nm). The morphology of the films was characterised using transmission electron microscopy (TEM; JEOL 3011, Japan), scanning transmission electron microscopy (STEM; Hitachi HD-2300A, Japan), and high-resolution TEM (HRTEM; Hitachi HD-3010A, Japan). Elemental compositions were determined using energy-dispersive X-ray spectroscopy (EDS; Oxford Instruments, UK) and inductively coupled plasma mass spectrometry (ICP-MS; Thermo Scientific XSeries 2 ICPMS, USA). The catalyst surface area was determined using Brunauer–Emmet–Teller (BET) analysis, using a BELSORP-mini II (BEL. Japan Inc.) under a flow of $N_2$ gas.

**OER characterisation.** OER characterisation was performed on a glassy carbon rotating disk electrode. First, 2 mg of catalyst was dissolved in 2 mL ethanol and 100 μL Nafion solution was added. Then, the mixture was sonicated for 30 min to form a homogenous mixture. Subsequently, 90 μL of the slurry was loaded onto the surface of a glassy carbon electrode (GCE; 0.196 $cm^2$) and the electrode was dried at room temperature. The electrolyte was purified to remove trace Fe using Ni(OH)₂ powder. The OER measurements were performed using a Voltalab PGZ-301 potentiostat/galvanostat (Radiometer Analytical, France), with a Pt foil and a Ag/AgCl electrode used as the counter and reference electrodes, respectively. The loading amount of the catalysts was 0.168 mg $cm^{-2}$. All potentials were plotted versus the reversible hydrogen electrode (RHE) as $E_{(RHE)} = E_{(Ag/AgCl)} + 0.197 + 0.0591 \times pH$. All linear sweep voltammetry measurements were performed at a scan rate of 5 mV $s^{-1}$. All OER measurements were iR-compensated (98%). Each measurement was conducted three times under the same conditions. The error bars denote variations observed from sample synthesis and OER measurements. The stability test was performed using the controlled current electrolysis method. PXRD measurements verified that all the obtained materials had the perovskite structure.

To evaluate the intrinsic activities, the current densities were normalised by the loading amount and the BET surface areas in order to exclude the increase in current as a result of high loading content and higher surface area. Normalisation was performed according to the expression: $i$ (mA $cm^{-2}$ oxide current) = $i$ (mA $cm^{-2}$ disk current) ÷ (loading amount (g $cm^{-2}$) × BET surface area ($cm^2$ $g^{-1}$)). Here, $i$ (mA $cm^{-2}$ oxide current) was denoted as the normalised specific activity, while $i$ (mA $g^{-1}$ oxide current) = $i$ (mA $cm^{-2}$ disk current) ÷ (loading amount (g $cm^{-2}$)) refers to the mass activity.

## Data availability
The data of measured $V_{RHE}$ values for all oxide perovskites and 3545 potential oxide perovskites listed by the amount of $\mu/t$ are provided online.

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

---

**Table 3 The setup of hyper-parameters in gplearn for GPSR.**

| Parameter | Value |
|---|---|
| population size | 5000 |
| Generations | 20 |
| stopping criteria | 0.01 (eV) |
| pc | 0.5, 0.95 (step = 0.025) |
| ps | (1-pc)/3, (0.92-pc)/3 (step = 0.01) |
| ph | ps |
| pp | 1-pc-ps-ph |
| function set | add, sub, mul, div, sqrt |
| parsimony coefficient | 0.0005, 0.0015 (step = 0.0005) |
| tournament size | 20 |
| metric | mean absolute error (MAE) |
| constant range | (−1,1) |

The explanation of each hyperparameter are as follows: *population size* the number of mathematical formulas in each generation, *generations* the max number of generations, *stopping criteria* MAE value that the program stops, *pc* crossover probability, *ps* subtree mutation probability, *ph* hoist mutation probability, *pp* point mutation probability, *function set* basic building blocks containing mathematical operators, *parsimony coefficient* a constant that penalizes large individuals by adjusting their MAE to make them less favorable for selection, *tournament size* the number of individuals in each tournament, *metric* measures how well an individual fits, *constant range* the range of constants included in mathematical formula.

9. Suntivich, J., May, K. J., Gasteiger, H. A., Goodenough, J. B. & Shao-Horn, Y. A perovskite oxide optimized for oxygen evolution catalysis from molecular orbital principles. *Science* **334**, 1383–1385 (2011).

10. Wei, C. et al. Recommended practices and benchmark activity for hydrogen and oxygen electrocatalysis in water splitting and fuel cells. *Adv. Mater.* **31**, 1806296 (2019).

11. Vojvodic, A. & Nørskov, J. K. Chemistry: optimizing perovskites for the water-splitting reaction. *Science* **334**, 1355–1356 (2011).

12. She, Z. W. et al. Combining theory and experiment in electrocatalysis: Insights into materials design. *Science* **355**, eaad4998 (2017).

13. Yin, W. J. et al. Oxide perovskites, double perovskites and derivatives for electrocatalysis, photocatalysis, and photovoltaics. *Energy Environ. Sci.* **12**, 442–462 (2019).

14. Bradley, K., Giagloglou, K., Hayden, B. E., Jungius, H. & Vian, C. Reversible perovskite electrocatalysts for oxygen reduction/oxygen evolution. *Chem. Sci.* **10**, 4609–4617 (2019).

15. Retuerto, M. et al. $La_{1.5}Sr_{0.5}NiMn_{0.5}Ru_{0.5}O_6$ double perovskite with enhanced ORR/OER bifunctional catalytic activity. *ACS Appl. Mater. Interfaces* **11**, 21454–21464 (2019).

16. Bockris, J. O. The Electrocatalysis of oxygen evolution on perovskites. *J. Electrochem. Soc.* **131**, 290 (1984).

17. Man, I. C. et al. Universality in oxygen evolution electrocatalysis on oxide surfaces. *ChemCatChem* **3**, 1159–1165 (2011).

18. Hwang, J. et al. Perovskites in catalysis and electrocatalysis. *Science* **358**, 751–756 (2017).

19. Jacobs, R., Hwang, J., Shao-Horn, Y. & Morgan, D. Assessing correlations of perovskite catalytic performance with electronic structure descriptors. *Chem. Mater.* **31**, 785–797 (2019).

20. Haverkort, M. W. et al. Spin state transition in $LaCoO_3$ studied using soft X-ray absorption spectroscopy and magnetic circular dichroism. *Phys. Rev. Lett.* **97**, 38–41 (2006).

21. Ghiringhelli, L. M., Vybiral, J., Levchenko, S. V., Draxl, C. & Scheffler, M. Big data of materials science: critical role of the descriptor. *Phys. Rev. Lett.* **114**, 105503 (2015).

22. Vladislavleva, E. Model-based problem solving through symbolic regression via pareto genetic programming. 169–172 (CentER, Center for Economic Research, 2008).

23. Bartel, C. J. et al. New tolerance factor to predict the stability of perovskite oxides and halides. *Sci. Adv.* **5**, eaav0693 (2019).

24. Meadowcroft, D. B. Low-cost oxygen electrode material. *Nature* **226**, 847–848 (1970).

25. Hong, W. T., Welsch, R. E. & Shao-Horn, Y. Descriptors of oxygen-evolution activity for oxides: a statistical evaluation. *J. Phys. Chem. C.* **120**, 78–86 (2016).

26. Davies, D. W., Butler, K. T. & Walsh, A. Data-driven discovery of photoactive quaternary oxides using first-principles machine learning. *Chem. Mater.* **31**, 7221–7230 (2019).

27. Lu, S. et al. Accelerated discovery of stable lead-free hybrid organic-inorganic perovskites via machine learning. *Nat. Commun.* **9**, 3405 (2018).

28. Li, Z., Xu, Q., Sun, Q., Hou, Z. & Yin, W. J. Thermodynamic stability landscape of halide double perovskites via high-throughput computing and machine learning. *Adv. Funct. Mater.* **29**, 1807280 (2019).

29. Stephens, T. gplearn. https://gplearn.readthedocs.io/en/latest/intro.html.

30. Petrie, J. R. et al. Enhanced bifunctional oxygen catalysis in strained $LaNiO_3$ perovskites. *J. Am. Chem. Soc.* **138**, 2488–2491 (2016).

31. Forslund, R. P. et al. Exceptional electrocatalytic oxygen evolution via tunable charge transfer interactions in $La_{0.5}Sr_{1.5}Ni_{1-x}Fe_xO_{4\pm\delta}$ Ruddlesden-Popper oxides. *Nat. Commun.* **9**, 3150 (2018).

32. Hona, R. K. & Ramezanipour, F. Remarkable oxygen-evolution activity of a perovskite oxide from the $Ca_{2-x}Sr_xFe_2O_{6-\delta}$ series. *Angew. Chem.* **131**, 2082–2085 (2019).

33. Weng, B. et al. A layered $Na_{1-x}Ni_yFe_{1-y}O_2$ double oxide oxygen evolution reaction electrocatalyst for highly efficient water-splitting. *Energy Environ. Sci.* **10**, 121–128 (2017).

34. Rong, X., Parolin, J. & Kolpak, A. M. A fundamental relationship between reaction mechanism and stability in metal oxide catalysts for oxygen evolution. *ACS Catal.* **6**, 1153–1158 (2016).

35. Feng, L. M. et al. Formability of $ABO_3$ cubic perovskites. *J. Phys. Chem. Solids* **69**, 967–974 (2008).

## Acknowledgements
W.Y. acknowledges funding support from the National Key Research and Development Program of China (grant No. 2016YFB0700700); National Natural Science Foundation of China (grant No. 11674237, 11974257) and the Priority Academic Program Development of Jiangsu Higher Education Institutions (PAPD). The theoretical work was carried out at the National Supercomputer Center in Tianjin and the calculations were performed on TianHe-1(A). This paper presents results from an NSF project (grant No. CBET−1433401) under the 'NSF 14−15: NSF/DOE Partnership on Advanced Frontiers in Renewable Hydrogen Fuel Production via Solar Water Splitting Technologies' project, which was co-sponsored by the National Science Foundation, Division of Chemical, Bioengineering, Environmental, and Transport Systems (CBET), and the U.S. Department of Energy, Office of Energy Efficiency and Renewable Energy, Fuel Cell Technologies Office.

## Author contributions
W.Y. and Y.Y. conceived the idea and supervised the project. W.Y. supervised the theoretical study. Y.Y. supervised the experimental study. W.Y., Y.Y., and Q.S. performed data collection and analysis. Z.S and W.Y. carried out SR calculations and analysis. B.W., C.G., Z.Z., and Q.Y. performed materials synthesis and OER characterisation. W.Y., Y.Y., and B.W. wrote the paper.

## Competing interests
The authors declare no competing interests.
