## [Peer Review File · Nature Communications]

REVIEWER COMMENTS

Reviewer #1 (Remarks to the Author):

The authors apply an existing symbolic regression (SR) technique, *gplearn*, to the problem of predicting oxygen evolution reaction (OER) activity for perovskite oxides given their composition. Using this approach, they identified that the ratio of octahedral factor (μ) to Goldschmidt's tolerance factor (t) (μ/t) is predictive of OER activity. To obtain this descriptor, the authors synthesized 18 perovskite oxides and measured their activity (VRHE) at various current densities. The performance of the descriptor was further validated by comparison to previous works and to 5 new perovskite oxides suggested by the model to have high activity. Overall, this work is a nice demonstration of how symbolic techniques (and the resulting "simple" descriptors) can be used to accelerate materials discovery. The experimental work to train and validate the model is impressive, as is the attempt to assess their model against known materials published in the literature. I'm supportive of its publication in *Nature Communications* after the following revisions are carefully considered.

1. The discussion of symbolic regression (SR) is a bit confusing and not properly contextualized with respect to its application to other similar problems and with respect to how it differs from what the authors described as "statistical ML". A few specific points follow :

a) The authors state that "Symbolic regression (SR) is an alternative approach that can search for an optimal function with multiple features as variables that describes a given dataset^{1,6}." This same description could be applied to any number of machine learning techniques – e.g., linear regression finds an optimal (linear) function of multiple features to describe a given dataset. The authors should clarify how SR (and specifically their application of SR) differs technically from other ML techniques.

b) The description of non-symbolic ML techniques as "statistical" is not quite correct. Of course, symbolic regression is also a "statistical" approach. The contrast between symbolic and non-symbolic techniques should be clarified.

c) It is incorrect to say that all non-symbolic ML techniques are "black-box" methods. There is an entire field of research on "interpretable ML" that pursues the extraction of meaning (in this case physical insights) from ML models that may or may not be symbolically based.

d) The authors state that "The good reliability and comparability of datasets used in SR analysis are of crucial importance for SR in order to produce accurate and insightful formulas^{19,20}". References 19 and 20 are general reviews on ML in materials science and do not make direct mention of symbolic regression, so their citation here is unclear. There are several examples of symbolic ML working with limited but consistent datasets that could be cited (e.g., applications of the SISO algorithm cited in this work often function with small curated datasets – one example is the highly relevant work on perovskite stability: Bartel et al. *Sci Adv* 2019 10.1126/sciadv.aav0693).

e) The authors state that "SR initially builds a population of random formulas with these parameters—normalised in advance to account for different dimensionalities—as variables." It is important to distinguish between what is a general aspect of SR and what is specific to the present application of SR. This sentence reads as though it is always true for any application of SR, but this is not necessarily the case.

2. It appears that μ/t is a necessary but not sufficient description of OER activity. For example, there are a number of ABO₃ perovskite that have $\mu/t < 0.4$ that I suspect will not be active for OER (though I could be wrong). A few compounds that come to mind are: LaAlO₃, CaSiO₃, SrGeO₃. Could the authors provide more guidance on how the descriptor μ/t should be applied and what other implicit criteria might be required to identify OER active perovskite catalysts?

a) Along these lines, it should be noted that the scope of materials investigated is actually quite small. For

example, in the training set, t spans only from 0.993 to 1.119 and μ spans only from 0.385 to 0.430. Perovskites are known to be stable at much lower t values (down to ~ 0.8) and much larger μ values (many at $\mu > 0.5$). Similarly, all materials investigated have reasonably high OER activity (whereas, of course, most compounds in the broad space of perovskites are inactive for OER). What is the generalizability outside of these small ranges?

3. The authors make several allusions to extracting “clear, physical insights” from their found descriptor(s). While it’s clear that μ/t is predictive of the set of materials analyzed, it’s not clear why this is the case. A low μ/t is correlated with good OER activity. As mentioned by the authors, this can be achieved by decreasing the radius of the B site or increasing the radius of the A site. However, it’s not clear why these radii changes dictate the OER activity. What are the chemical implications of having a large r_A/r_B ratio on catalytic activity?

a) The coefficients in the expression for VRHE vs μ/t often change quite a bit. That is, the slope and intercept of $\text{VRHE} = (\text{slope}) * \mu/t + \text{intercept}$ are not consistent throughout this work. For example, in Figure 3b, the slope appears to be same as reported in Table 2 (slope = 1.554). The slope (and intercept) change for the plotting of Figure 3d – by inspection the slope looks to be ~ 2 , and it is certainly steeper than 1.554. There are even more extreme examples in the SI – the slope appears to be ~ 1 in Fig S2 and varies dramatically through Figures S3 and S4. 1) the authors should provide the slope and intercept for the best fit on each of these panels and 2) the authors should provide a physical explanation for why the coefficients of the found descriptor are so variable with respect to data set. This variability suggests a lack of generalizability of the found descriptor (μ/t) unless it is grounded in some physical explanation.

b) It is similarly important to note that while minimizing μ/t is found to maximize OER activity in this work, doing so should generally have a negative effect on stability. The materials studied in this work already start from a point of having a large r_A/r_B ratio. That is, their t values of ~ 1 are on the large side and their μ values of ~ 0.4 are on the small side. Further increases in t and decreases in μ will typically destabilize the perovskite structure. Some discussion of stability would be prudent.

4. The motivation for using t and μ (which themselves are constructed symbolic expressions) as input features is not clear. Is it necessary or even helpful to use these as a starting point?

a) Is it possible to discover an even simpler or more predictive descriptor if r_A , r_B , and r_X are used as input features and t (μ) is not inputted a priori? Will the algorithm “find” t (μ) on its own? μ/t is analyzed from the perspective of increasing r_A and decreasing r_B – is it possible that r_A/r_B is actually a similarly good or perhaps even better descriptor?

b) Moreover, is r_X even necessary? The inclusion of r_X in μ and t is irrelevant to the present work as $X = O$ for all compounds studied. Shouldn’t a simpler and more intuitive descriptor be discoverable if the descriptor is built only on the properties of A and B (which are varied in this work)?

A few minor points:

1. The statement that “Recent research shows that about twenty samples are good for finding a trend to accelerate the design of inorganic materials” is extremely system-, data-, and algorithm-dependent and is by no means a general rule of thumb. There are several ML problems in materials science that require many thousands of examples. While it is clear that in the cited works and in the present paper, it is possible to find predictive descriptors from < 20 data points, it is misleading to suggest that this finding is general to “the design of inorganic materials”. This is more a function of how focused the present work is on a very small subset of inorganic materials – perovskite oxides with large A-site cations that have reasonably high OER activities.

2. Regarding descriptor identification and "complexity", how was the inherent complexity of t and μ factored in? That is, t is already a relatively complex function of r_A , r_B , and r_X . Was this considered in the "complexity" metric?
3. The use of 1080 data points generated for 18 unique materials is puzzling. The use of 4 different samples and 3 different measurements appears to be only a matter of data replication (i.e., if there is no noise in the sample preparation or measurement, then these 12 data points should be identical). Why not just use the mean value from these 12 experiments?
4. I gather that varying the current density has an effect on the obtained OER activity. Can the authors comment more on the effect of sampling different current densities for training the descriptor?
5. Regarding the prediction of thousands of new candidate OER catalysts, the statement that "Subject to the requirement of charge balance, 3,545 oxide perovskites were obtained from the SR analysis" is unclear. SR produces the descriptor μ/t , then these 3545 compounds were found to have low μ/t . SR does not produce the formulas directly as suggested by this sentence.
6. Some comments on the potential stability of these 3545 compounds would also be prudent.

Reviewer #2 (Remarks to the Author):

In this manuscript, Weng et al. apply symbolic regression to guide the design of new oxide perovskites for the OER. The authors provide an interesting activity descriptor for perovskite OER catalysts, which is based on the fraction between the octahedral factor and the tolerance factor. However, serious concerns on the manuscript persist in terms of scientific presentation and reproducibility of scientific results.

More information is needed in the introductory part about recent achievements on perovskites with high activity toward OER. For instance, the authors simply cite the review in Ref. 11, but a brief explicit discussion of the main results is important for the reader. Recently, Bradley et al. (DOI:10.1039/c9sc00412b) and Retuerto et al. (DOI:10.1021/acscami.9b02077) proposed similar perovskites for the OER, which might be worth to mention.

Entire parts of the manuscript are written imprecisely, and important information is missing or remains vague. For example, the following phrases need to be clarified precisely in the manuscript:

- 1) which "meaningful descriptors" are the authors referring to in line 60?
- 2) what is the "simple and accurate descriptor" in line 60?
- 3) what are "clear physical insights" in line 60?
- 4) what kind of "strategies" do the descriptors "help develop" (see line 61)?
- 5) what does the phrase "high-throughput screening" mean precisely (see line 63)? A concrete description is needed.
- 6) in what sense the "new oxide perovskites" (see line 65) are new? Have they never been synthesized before? Have they never been suggested as active OER catalysts before?
- 7) what is a "potentially high OER activity"? Does the word "potential" refer to predicted activities, which have not been confirmed by experiments, or do the authors target a different meaning?
- 8) in what sense the datasets used in SR analysis have a "good reliability" and "comparability" (see line 68)? "Reliability" and "comparability" will be proper quantities if and only if they are measured quantitatively. How do the authors measure "reliability" and "comparability" quantitatively in line 68?
- 9) in what sense the formulas mentioned in line 69 are "accurate"? Again, "accurate" needs a quantitative measure.
- 10) what is the precise meaning of the word "insightful" in the phrase "insightful formulas" (see line 69)?
- 11) what is "structural flexibility" and what is "chemical flexibility" of perovskite structures stated in line 71? Do the authors refer to the fact that perovskites can take several different lattice structures (classified by symmetry groups) and that several different material compounds are called perovskites?

12) what are "obvious physical insights into the studied data sets" referred to in line 32?

13) what is "the most parsimonious way" in line 98?

The authors' statement in the first paragraph that "statistical ML does not provide obvious physical insights into the studied data sets" is dubious:

There is simply no need to apply any statistical ML or any other advanced method to provide "obvious" physical insights. If neglected obvious, this statement will be presumptuous in the sense that it disregards the variety of recent achievements by statistical ML in materials science.

The authors state that "symbolic regression is a unique machine learning approach". Thus, how does the authors' machine learning approach differ from statistical ML, which the authors try to avoid?

The authors' statement that the symbolic regression "is different from statistical machine-learning approach, which bears a hidden black-box model and is difficult for physical interpretation" is not a reasonable argument and in particular, this is not a scientific argument publishable in Nature Communications!

The authors claim that "statistical ML is not easily accessed by many researchers, particularly by experimentalists". The authors further state that SR is "in contrast to statistical ML, which is a 'black-box'". However, the authors do not clarify that SR is more easily accessible than statistical ML. What do the authors mean exactly by the "black-box"?

Given that, for instance, statistical ML using Kernel Ridge Regression is based on a supervised least-square fit with penalty term, statistical ML is not generally a "black-box". Therefore, the authors' statement in the manuscript's first paragraph is questionable.

The authors use the orbital filling, but without providing any definition of this quantity. How is the orbital filling mathematically defined in this study and how is it calculated in this study? Which orbitals are the authors referring to?

What are the physical (or geometrical) interpretations of the "octahedral" and "tolerance" factors, respectively? In the definition in line 94, the quantities r_A , r_B , r_O are changed to R_A , R_B , R_O in the Supporting Information without reasoning. To be precise, I suggest to provide the definition of the tolerance and octahedral factors as equations. The expression $t(\dots)$ is usually read as "t is a function of".

How are "weighted averaged ionic radii" in R_A and R_B defined? Providing all mathematical definitions is mandatory to make results reproducible.

In Fig. 2a, what is the precise meaning of "materials index"? What information and experiments include the "no. of experiments" in the plot in Fig. 2a? The quantity "no. of experiments" seems to not been mentioned anywhere in the main text.

In Fig. 3c and in line 46, what does e_g refer to?

Figures in the Supporting Information (and in the main text) should appear within the paragraphs, in which they are referenced.

Symbolic regression (SR) is the central method in the theory part of the manuscript. However, SR is not described comprehensively and not presented appropriately: For details on SR, in the main text the authors refer to "Fig. 2b and Supplementary Information". In Fig. 2b, the authors again refer to the Supplementary Information for details on SR. In the brief SR paragraph in the Supplementary Information, the authors in turn refer to the "Method part" in the main text. In the Method part in the main text, the authors in turn refer to Tab. S4 in the Supporting Information. At the end of the brief Method part about SR in the main text, the authors state that "the details of the symbolic regression approach are provided as Supplementary Information". Hence, this presentation of the SR method is confusing. Thus, I propose to include all relevant information about SR, which are needed to enable the reader to understand the SR comprehensively without additional sources, in the Method part in the main text.

Regarding the SR method description and the associated flowchart in Fig. 2b :

- 1) What are the "432 sets of hyper-parameters"? How large are the sets? What are the hyperparameters? What do is the effect of the hyperparameters?
- 2) What are the "1000 selected formulas"? What do the formulas describe? Does "formula" (see also line 69) refer to mathematical or chemical formulas? How are the formulas generated or selected? This all remains absolutely unclear.
- 3) What are "random analytical formulas"? What means "random" here, what means "analytical" here, and (see point above) what means "formula" here?
- 4) What are "random analytical forms" in Fig. 2b? How are they "created"? Do the authors use "forms" as synonym for "formulas" and, if yes, why?
- 5) How are "random tree structures" generated? Associated mathematical details of the method should be explained in the manuscript.
- 6) How is the "performance of formula [...] measured by mean absolute errors"? What is the precise meaning of "performance" in the context of "formula"?
- 7) What is a "final solutions set"? What means "solution" here?
- 8) How to "form 1000 new" "generic operations of crossover and mutation"? What means "generic" here?
- 9) What is the "fitness of each form" (see Fig. 2b)? How is "fitness" measured quantitatively and mathematically?
- 10) What is the "termination criteria" (see Fig. 2b)?
- 11) What are the "genetic operations" applied in Fig. 2b? Does Fig. 2b here refer to "crossover and mutation", which are mentioned in the Supporting Information?
- 12) How are "descendants" added as stated in Fig. 2b? How are those "descendants" generated?
- 13) In line 36, what are the "analytical forms that may have physical meanings"? What "physical meanings" do the authors refer to exactly?
- 14) Regarding symbolic regression, why is it called "symbolic"? In what sense it is a "regression"?
- 15) What is the link between symbolic regression and the therein described genetic algorithm?
- 16) Vague explanations in the manuscript such as "symbolic regression is a unique machine learning approach" are not helpful for understanding the method. What do the authors mean by "unique"?
- 17) What are "best forms" stated in Fig. 2b? What does "best" refer to and how are "best forms" determined quantitatively? Is the activity of the perovskites determined in the theoretical approach and, if yes, how?

Since the SR method is not well described, it is not reproducible how the resulting perovskites are selected. Therefore, I suggest to present the theoretical method not in a general scheme, but in terms of how the perovskite structures are generated, evaluated, and optimized.

The authors state that the suggested perovskites are "among the oxide perovskite catalysts with the highest specific activities (Ref. 8)". This is imprecise: Which perovskites from their studies do the authors compare to which perovskites in literature? What are the activities of both? Are all activities measured in the same experimental environment, and are the activities comparable and, if yes, why?

In Fig. 3d, the octahedral/tolerance descriptor is applied on perovskites from literature. How does the descriptor describe the perovskites from this study? Why is this descriptor, which is even highlighted in the abstract, applied only on literature data, but without applying the descriptor on the data from this study?

In sum, regarding the above-discussed points I cannot recommend this manuscript for publication, in particular with regard to Nature Communications.

Reviewer #3 (Remarks to the Author):

I previously reviewed this manuscript for another NPG journal. The revision has greatly improved it and addressed my previous concerns well.

Weng et al. report an attractively simple structural descriptor for the oxygen evolution reaction of perovskite

oxides, namely μ/t . The descriptor is identified among others possible choices by symbolic regression where simplicity is a criterion for selection. The authors use it to predict new perovskite catalysts that are synthesized, phase checked by XRD and then tested by cyclic voltammetry and chronoamperometry.

The main claims are that the work is based on the new descriptor μ/t and that the new catalysts are superior to BSCF. Both claims are well supported and the work is quite impressive due to the rational catalyst design and verification of the high activity of the newly predicted electrocatalysts. Moreover, the approach to identification of new descriptors is very promising. I believe the simple descriptor and the novel predictive approach are outstanding features of the work and they will be important for rational design of future perovskite catalysts in the electrocatalysis community as well as related catalysis communities where perovskites are used.

In this version, the statistical foundation of the identification of the new descriptor is easier to follow for non-experts.

The proposed descriptor μ/t is indeed new and very useful. It is limited to perovskite and perovskite-like structures, which are important materials classes for catalysis. The authors calculate μ/t for several current densities in their own systematic dataset and also for several previously reported systematic studies where the activity trends are correctly predicted. To test the generality of the new simple descriptor, I also asked my team to plot the OER activities of our literature database against μ/t obtained from Shannon's table where again the reported activities are anti-proportional to μ/t . Thus, I am confident that this descriptor will be used even though many others (such as eg occupancy) have been proposed before. It is superior to many other descriptors because tabulated values of the ion radii can be used which removes some ambiguity of obtaining the spin state and valence (eg occupancy) or calculating reaction free energies by DFT.

The authors rationalize the descriptor based on cation sizes and the perovskite stability. It is recommended to look for new catalysts at the boundary where the perovskite structure is predicted to become unstable (according to Goldschmidt's rules) but still adopts the perovskite experimentally. Monovalent A-sites are currently rarely used in perovskite oxide catalysts but the descriptor μ/t uniquely suggests their attractiveness and they may become more common due to the authors' work.

I was also very impressed that the authors not only predict a new descriptor but also systematically synthesize new perovskite oxides with new compositions and test their OER activity. The determined intrinsic activities of the new perovskite oxides rank among the best reported so far.

Minor points:

- 1) The information in the methods sections "OER characterization" and "electrochemical characterization" seems to be mainly duplicated. Please merge them.
- 2) The eg descriptor is commonly attributed to the original research publication (Suntivich, Science 2011; ref. 7) and not later reviews of the Shao-Horn group. Ref. 14 could still be cited for an updated discussion.
- 3) Fig. 2a is very difficult to understand as well as crowded and thus unclear. Can it be plotted in an x-y plot, e.g. by using the average and standard deviation rather than experiment # as an axis? The data are also submitted as excel sheets where one can look up the individual values (which I find commendable).

I support the publication of this manuscript in Nature Communications. The minor points could be addressed without further review.

Marcel Risch

REVIEWER COMMENTS

Reviewer #1 (Remarks to the Author):

The authors apply an existing symbolic regression (SR) technique, `gplearn`, to the problem of predicting oxygen evolution reaction (OER) activity for perovskite oxides given their composition. Using this approach, they identified that the ratio of octahedral factor (μ) to Goldschmidt's tolerance factor (t) (μ/t) is predictive of OER activity. To obtain this descriptor, the authors synthesized 18 perovskite oxides and measured their activity (VRHE) at various current densities. The performance of the descriptor was further validated by comparison to previous works and to 5 new perovskite oxides suggested by the model to have high activity. Overall, this work is a nice demonstration of how symbolic techniques (and the resulting "simple" descriptors) can be used to accelerate materials discovery. The experimental work to train and validate the model is impressive, as is the attempt to assess their model against known materials published in the literature. I'm supportive of its publication in Nature Communications after the following revisions are carefully considered.

Response: We thank the Reviewer for his/her careful reading of our manuscript and support of our work.

1. The discussion of symbolic regression (SR) is a bit confusing and not properly contextualized with respect to its application to other similar problems and with respect to how it differs from what the authors described as "statistical ML". A few specific points follow:

Response: We thank the reviewer for these comments. As far as we know, in machine learning (ML) fields, there is still ambiguity in the relationship and difference among the terminologies among 'statistics', 'machine learning', 'statistical learning', 'statistical machine learning', and 'symbolic regression (SR)'. For example, Wang *et al.* consider SR as an alternative method to ML [MRS Comm. 9, 793 (2019)], while Jin *et al.* consider SR as a kind of interpretable ML approach [arXiv:1910.08892v3 (2020)]. Meanwhile, although **Fig. R1** shows the relation and difference between 'statistics', 'machine learning' and 'statistical learning' [<https://www.datasciencecentral.com/profiles/blogs/machine-learning-vs-statistics-in-one-picture>], 'statistical machine learning' is sometimes quoted as synonyms for 'machine learning' [*Introduction to Statistical Machine Learning* by Masashi Sugiyama, Elsevier (2016); *An Introduction to Statistical Learning* by G. James, D. Witten, T. Hastie and R. Tibshirani, Springer (2017)]. To avoid confusion, in our revision, we modified the statement by considering SR as a kind of interpretable ML and discarding the word of 'statistical ML'.

Musio image: Akawikipic [CC BY-SA 4.0 (<https://creativecommons.org/licenses/by-sa/4.0/>)]

Fig. R1 The relation and difference between ‘statistics’, ‘machine learning’ and ‘statistical learning’.

The point-by-point responses are following:

a) The authors state that “Symbolic regression (SR) is an alternative approach that can search for an optimal function with multiple features as variables that describes a given dataset^{1,6}.” This same description could be applied to any number of machine learning techniques – e.g., linear regression finds an optimal (linear) function of multiple features to describe a given dataset. The authors should clarify how SR (and specifically their application of SR) differs technically from other ML techniques.

Response: SR simultaneously searches for the optimal formula of a function and set of parameters in the function. Nevertheless, linear regression (or other similar regression approach) fixed the function model (for example, linear function) and fit the parameters. In revision, we modified the statement to “Symbolic regression (SR) is an approach of interpretable machine learning that simultaneously searches for the optimal mathematical formula of a function and set of parameters in the function.”

b) The description of non-symbolic ML techniques as “statistical” is not quite correct. Of course, symbolic regression is also a “statistical” approach. The contrast between symbolic and non-symbolic techniques should be clarified.

Response: The term of ‘statistical ML’ is adopted from the book of ‘*Introduction to Statistical Machine Learning*’ by Masashi Sugiyama, Elsevier (2016). We agree with the reviewer that in principle, SR itself is a statistical approach. In the revision, we have discarded the word of ‘statistical ML’, used ‘ML’ instead and considered SR as a kind of interpretable ML.

c) It is incorrect to say that all non-symbolic ML techniques are “black-box” methods. There is an entire field of research on “interpretable ML” that pursues the extraction of meaning (in this case physical insights) from ML models that may or may not be symbolically based.

Response: The reviewer is correct. In the revision, we considered SR as a kind of interpretable ML method [arXiv:1910.08892v3 (2020)] and modified the sentence to “SR is capable to deliver interpretable mathematical formulas that may provide direct guidance for materials design”.

d) The authors state that “The good reliability and comparability of datasets used in SR analysis are of crucial importance for SR in order to produce accurate and insightful formulas^{19,20}”. References 19 and 20 are general reviews on ML in materials science and do not make direct mention of symbolic regression, so their citation here is unclear. There are several examples of symbolic ML working with limited but consistent datasets that could be cited (e.g., applications of the SISO algorithm cited in this work often function with small curated datasets – one example is the highly relevant work on perovskite stability: Bartel et al. Sci Adv 2019 10.1126/sciadv.aav0693).

Response: We thank referee for pointing out this. The relevant reference is now cited in revision.

e) The authors state that “SR initially builds a population of random formulas with these parameters—normalised in advance to account for different dimensionalities—as variables.” It is important to distinguish between what is a general aspect of SR and what is specific to the present application of SR. This sentence reads as though it is always true for any application of SR, but this is not necessarily the case.

Response: We modified the statement to “In this work, SR initially builds a population of random mathematical formulas with these parameters as variables”.

2. It appears that μ/t is a necessary but not sufficient description of OER activity. For example, there are a number of ABO₃ perovskite that have $\mu/t < 0.4$ that I suspect will not be active for OER (though I could be wrong). A few compounds that come to mind are: LaAlO₃, CaSiO₃, SrGeO₃. Could the authors provide more guidance on how the descriptor μ/t should be applied and what other implicit criteria might be required to identify OER active perovskite catalysts?

Response: We thank the reviewer for this insightful comment. We did calculate μ/t for LaAlO₃ (0.390), CaSiO₃ (0.272) and SrGeO₃ (0.374). Among them, CaSiO₃ was predicted to have very high OER activity according to μ/t values. We searched literature and did not find any experimental report of catalysis application for those materials. There are two possibilities: (i) those compounds may have catalysis activity and deserve to be studied; (ii) in this work, all oxide perovskite samples in both training and test set contain transition metal elements on B sites, therefore, the μ/t may only apply to oxide perovskites with transition metal elements on B sites.

a) Along these lines, it should be noted that the scope of materials investigated is actually quite small. For example, in the training set, t spans only from 0.993 to 1.119 and μ spans only from 0.385 to 0.430. Perovskites are known to be stable at much lower t values (down to ~ 0.8) and much larger μ values (many at $\mu > 0.5$). Similarly, all materials investigated have reasonably high OER activity (whereas, of course, most compounds in the broad space of

perovskites are inactive for OER). What is the generalizability outside of these small ranges?

Response: Inspired by referee's comment, we exhaustively searched ICSD (Inorganic Crystal Structure Database) and found that the existing oxide perovskites mostly have $t < 0.95$ and $\mu > 0.55$ (Fig. R2). However, oxide perovskites reported to be catalyst in the last forty years lie in a small confined range [$t > 0.95$ and $\mu < 0.55$]. According to μ/t , most of oxide perovskites are less catalytically active, which seems consistent with existing experimental results that oxide perovskite catalysts are limited in a few types of perovskites [Adv. Mater. 31, 1806296 (2019)].

Fig. R2 The t - μ map of 534 perovskites (blue circle) with ABO_3 and $A_2B'B''O_6$ formula found in ICSD. The eighteen training samples are marked as orange triangle and five new perovskites in this work are marked as red star.

This is indeed a direction deserving further research. We will continue our research and hope current work would stimulate upcoming work to verify the generality of this descriptor and to explore more oxide perovskites for catalysis applications.

3. The authors make several allusions to extracting “clear, physical insights” from their found descriptor(s). While it's clear that μ/t is predictive of the set of materials analyzed, it's not clear why this is the case. A low μ/t is correlated with good OER activity. As mentioned by the authors, this can be achieved by decreasing the radius of the B site or increasing the radius of the A site. However, it's not clear why these radii changes dictate the OER activity. What are the chemical implications of having a large r_A/r_B ratio on catalytic activity?

Response: We revised our manuscript by deleting the statement of ‘clear, physical insights’ in revision. So far, we do not understand the underlying physics why large r_A and small r_B lead to high chemical activity. It may imply that the catalytic activity is closely related to the structural stability, *i.e.* a lower stability leads to a high activity. In revision, we added statement before conclusion part on P5 that “More in-depth understanding of correlation among μ/t , catalysis activity and structural stability is out of scope of current research but deserves further study.” We hope current work may stimulate the fundamental research along this direction.

a) The coefficients in the expression for VRHE vs μ/t often change quite a bit. That is, the slope and intercept of $VRHE = (\text{slope}) * \mu/t + \text{intercept}$ are not consistent throughout this work. For

example, in Figure 3b, the slope appears to be the same as reported in Table 2 (slope = 1.554). The slope (and intercept) change for the plotting of Figure 3d – by inspection the slope looks to be ~2, and it is certainly steeper than 1.554. There are even more extreme examples in the SI – the slope appears to be ~1 in Fig S2 and varies dramatically through Figures S3 and S4. 1) the authors should provide the slope and intercept for the best fit on each of these panels and 2) the authors should provide a physical explanation for why the coefficients of the found descriptor are so variable with respect to data set. This variability suggests a lack of generalizability of the found descriptor (μ/t) unless it is grounded in some physical explanation.

Response: The linear function for the best fit on related plots have been shown in **Fig. R3** below. Their slopes and intercepts vary, because the V_{RHE} values are sensitive to specific experimental and measurement conditions, including loading amount, surface areas and measurements methods (intrinsic activity or Tafel slopes), therefore, the experimental V_{RHE} values can vary by up to 0.2 eV from different experiments (Adv. Mater. 2015, 27, 266; Electro. Acta 2017, 241, 433). In this sense, the fitting slope and intercept for different experiments may differ a lot. However, the monotonic trends do not change. Therefore, μ/t was proposed as descriptor rather than the explicit mathematical formula with slope and intercept.

Fig. R3 The fitting functions in the related figures of manuscript.

Fig. S2 compared our measured V_{RHE} values (x-axis) with ones from previous literature [Science 334, 1383 (2011)] for the same oxide perovskites. The x-axis is not μ/t . Therefore, the slope is about 1, but not exactly 1 due to reasons mentioned above.

b) It is similarly important to note that while minimizing μ/t is found to maximize OER activity in this work, doing so should generally have a negative effect on stability. The materials studied in this work already start from a point of having a large r_A/r_B ratio. That is, their t values of ~ 1 are on the large side and their μ values of ~ 0.4 are on the small side. Further increases in t and decreases in μ will typically destabilize the perovskite structure. Some discussion of stability would be prudent.

Response: We thank referee for pointing out this important issue. The corresponding discussions are added in discussion part on P5, as following.

“The descriptor of μ/t implies that the catalytic activity of oxide perovskites is closely related to their structural stability, i.e. a lower stability leads to a high activity. Feature analysis in SR process shows that μ , t , and Q_A correlate with the catalytic activity more than R_A , N_d , χ_A , and χ_B (Fig. S9). The oxide perovskites showing improved OER activity had $t > 1$ (Tab. 1 and also Tab. 6 in Ref. 11), which were considered unstable perovskites³⁵. However, we found that these perovskites could be synthesised under suitable conditions. More in-depth understanding of correlation among μ/t , catalysis activity and structural stability is out of scope of current research but deserves further study.”

4. The motivation for using t and μ (which themselves are constructed symbolic expressions) as input features is not clear. Is it necessary or even helpful to use these as a starting point?

a) Is it possible to discover an even simpler or more predictive descriptor if r_A , r_B , and r_X are used as input features and t (μ) is not inputted a priori? Will the algorithm “find” t (μ) on its own? μ/t is analyzed from the perspective of increasing r_A and decreasing r_B – is it possible that r_A/r_B is actually a similarly good or perhaps even better descriptor?

Response: μ and t are chosen as the input feature since they are commonly-used features in machine-learning models of perovskite research [Nat. Comm. 2018, 9, 3405; Sci. Adv. 2019, 5, eaav0693; Adv. Fun. Mater. 2019, 29, 1807280; Chem. Mater. 2019, 31, 7221], although there are the functions of r_A , r_B and r_X . We followed referee’s suggestions and redo SR analysis based on the parameters of r_A , r_B , N_d , χ_A , χ_B , Q_A without t , μ . The results are shown in **Fig. R4** and **Table R1**. It is found that r_A/r_B is not on the Pareto front in this case. We also manually calculate MAE of r_A/r_B (0.274 meV), which performs worse than μ/t (MAE 0.253 meV). Unfortunately, SR did not find μ , t , or μ/t , probably due to the ergodic limitation of current genetic algorithm in functional space.

Fig. R4 Pareto front of SR based on parameters r_A , r_B , N_d , χ_A , χ_B , Q_A .

Table R1 The formulas on the Pareto front of Fig. R4.

Point	Formulas	MAE (eV)	Complexity
A	$0.162+r_B$	0.0333	1
B	$(1.455\chi_A+1.359)^{0.5}$	0.0322	2
C	$(Q_A^{0.5}+1.358)^{0.5}$	0.0256	3
D	$((Q_A+r_B)^{0.5}+1.157)^{0.5}$	0.0248	5
E	$(Q_A^{0.5}+(2.225r_A)^{0.25})^{0.5}$	0.0246	6
F	$\left(\frac{(1.733r_A)^{0.5}}{\chi_B^{0.25}}+Q_A^{0.5}\right)^{0.5}$	0.0235	9
G	$\left(\frac{((r_B+1.123)r_A)^{0.5}}{\chi_B^{0.25}}+Q_A^{0.5}\right)^{0.5}$	0.0232	11

b) Moreover, is rX even necessary? The inclusion of rX in mu and t is irrelevant to the present work as X = O for all compounds studied. Shouldn't a simpler and more intuitive descriptor be discoverable if the descriptor is built only on the properties of A and B (which are varied in this work)?

Response: We agree with referee that r_X is not necessary. Therefore, in our study, r_X was not considered as a feature. **Fig. R4** and **Tab. R1** is the SR results based on the parameters of r_A , r_B , N_d , χ_A , χ_B , Q_A . Unfortunately, compared to Table 2 in manuscript, the MAE of descriptors at the same complexity on Pareto front are mostly larger than the descriptors discovered based on μ , t , r_A , N_d , χ_A , χ_B , Q_A in our study.

A few minor points:

1. The statement that “Recent research shows that about twenty samples are good for finding a trend to accelerate the design of inorganic materials“ is extremely system-, data-, and algorithm-dependent and is by no means a general rule of thumb. There are several ML problems in materials science that require many thousands of examples. While it is clear that in the cited works and in the present paper, it is possible to find predictive descriptors from < 20 data points, it is misleading to suggest that this finding is general to “the design of inorganic materials”. This is more a function of how focused the present work is on a very small subset of inorganic materials – perovskite oxides with large A-site cations that have reasonably high OER activities.

Response: We agree with referee’s comments. To be precise, we deleted the sentence “Recent research shows that about twenty samples are good for finding a trend to accelerate the design of inorganic materials” in the revision.

2. Regarding descriptor identification and “complexity”, how was the inherent complexity of t and μ factored in? That is, t is already a relatively complex function of r_A , r_B , and r_X . Was this considered in the “complexity” metric?

Response: Since t or μ was considered as an input parameter in this work, its complexity is 1.

3. The use of 1080 data points generated for 18 unique materials is puzzling. The use of 4 different samples and 3 different measurements appears to be only a matter of data replication (i.e., if there is no noise in the sample preparation or measurement, then these 12 data points should be identical). Why not just use the mean value from these 12 experiments?

Response: The referee is correct that, ideally, these 12 data points should be identical. Nevertheless, as discussed above, V_{RHE} values are sensitive to specific experimental and measurement conditions, including loading amount, surface areas and measurements details, leading to the discrepancies for different samples and measurements. We considered all 12 data points, since we do not want to lose the information embedded in data distribution and discrepancies. Following referee’s idea, we used mean values to do SR. The results of Pareto front are shown in Fig. R5, which is very similar to former results. μ/t remains good balance between complexity and simplicity.

Fig. R5 Pareto front of SR using mean values of twelve experiments/measurements.

4. I gather that varying the current density has an effect on the obtained OER activity. Can the authors comment more on the effect of sampling different current densities for training the descriptor?

Response: Referee's observation is correct that V_{RHE} is dependent on current density, which is reflected as Tafel slope (Fig. 4b). The effect of descriptors on different current densities are shown in Fig. S3.

5. Regarding the prediction of thousands of new candidate OER catalysts, the statement that "Subject to the requirement of charge balance, 3,545 oxide perovskites were obtained from the SR analysis" is unclear. SR produces the descriptor μ/t , then these 3,545 compounds were found to have low μ/t . SR does not produce the formulas directly as suggested by this sentence.

Response: We thank referee for pointing out this and have deleted the words of "from the SR analysis".

6. Some comments on the potential stability of these 3,545 compounds would also be prudent.

Response: We modified the statement on P4 that "The formability and stabilities of 3,545 oxide perovskites have not been verified. Therefore, we selected thirteen new oxide perovskites ... for experimental verification ... We found that eight of them contained significant amounts of impurity or secondary phases ... Five compounds ... formed pure perovskite phases ...".

We thank the referee again for the nice comments and suggestions, which are very insightful and inspiring. We hope the referee will find his/her comments been properly addressed and the current revision is acceptable for publication.

Reviewer #2 (Remarks to the Author):

In this manuscript, Weng et al. apply symbolic regression to guide the design of new oxide perovskites for the OER. The authors provide an interesting activity descriptor for perovskite OER catalysts, which is based on the fraction between the octahedral factor and the tolerance factor. However, serious concerns on the manuscript persist in terms of scientific presentation and reproducibility of scientific results.

Response: We thank the referee for his/her interest and extremely careful reading of our manuscript. We appreciate referee's critical comments that help significantly improve the quality of our paper.

More information is needed in the introductory part about recent achievements on perovskites with high activity toward OER. For instance, the authors simply cite the review in Ref. 11, but a brief explicit discussion of the main results is important for the reader. Recently, Bradley et al. (DOI:10.1039/c9sc00412b) and Retuerto et al. (DOI:10.1021/acsami.9b02077) proposed similar perovskites for the OER, which might be worth to mention.

Response: We thank the referee for providing recent progress on oxide perovskite catalysts with bifunctional performance of OER/ORR. We added the statement "Moreover, oxide perovskites have recently been extended to the bifunctional application of OER and oxygen reduction reaction^{14,15}" in introduction part and the suggested reference have been properly cited (as Ref. [14] and [15] in revision).

Entire parts of the manuscript are written imprecisely, and important information is missing or remains vague. For example, the following phrases need to be clarified precisely in the manuscript:

1) which "meaningful descriptors" are the authors referring to in line 60?

Response: In original version, we want to state if the data were not comparable, the derived descriptor is meaningless. Considering this is a commonsense in machine learning (ML) field, in revision, we deleted this sentence to avoid confusion.

2) what is the "simple and accurate descriptor" in line 60?

Response: "Simple" and "accurate" are the two criteria to choose final descriptor. As shown in Fig. 3a, descriptor is more accurate when it has more complexity. Therefore, a good descriptor should be at the balance of simplicity and accuracy. In our specific study, we choose μ/t according to those criteria. To make it clear, the statement has been modified to "A descriptor with the balance of simplicity and accuracy is then chosen ...".

3) what are "clear physical insights" in line 60?

Response: In original version, we use “clear physical insight” for SR, in comparison to the “black-box” model of ML. We have deleted the words of "clear physical insights" in revision to soften our tone.

4) what kind of "strategies" do the descriptors "help develop" (see line 61)?

Response: The SR-derived descriptor μ/t indicates that oxide perovskite ABO_3 with larger cation A and smaller cation B should have higher catalysis activity. Accordingly, we design oxide perovskites by adopting large A cation and small B cation, as stated from line 122 to line 142 in original version.

5) what does the phrase "high-throughput screening" mean precisely (see line 63)? A concrete description is needed.

Response: High-throughput screening is a method of scientific experimentation or computation that comprises the screening of large number of compounds via the use of automation, miniaturized assays, and large-scale data analysis. In the original version, we used ‘high-throughput’ since the descriptor can complete fast screening among a library of 3,545 perovskites. To avoid confusion, ‘high-throughput screening’ in line 63 (original version) has been modified to ‘materials screening’ in revision.

6) in what sense the "new oxide perovskites" (see line 65) are new? Have they never been synthesized before? Have they never been suggested as active OER catalysts before?

Response: As far as we know, these five oxide perovskites have never been synthesized or suggested as OER catalyst. This is also confirmed by Referee #3.

7) what is a "potentially high OER activity"? Does the word "potential" refer to predicted activities, which have not been confirmed by experiments, or do the authors target a different meaning?

Response: The referee is correct that "potential" refers to predicted activities, which have not been confirmed by experiments. In this work, we chose thirteen potential oxide perovskites with high OER activity and successfully synthesized five ones (line 143 - 157 in original version).

8) in what sense the datasets used in SR analysis have a "good reliability" and "comparability" (see line 68)? "Reliability" and "comparability" will be proper quantities if and only if they are measured quantitatively. How do the authors measure "reliability" and "comparability" quantitatively in line 68?

Response: Here, "reliability" and "comparability" are not quantities that can be measured quantitatively but general statement to underline the data quality for SR. Immediately after that sentence, the V_{RHE} values from different groups and ages were summarized and found to be lack of comparability. For examples, the experimental V_{RHE} values can vary up to 0.2 eV from different

experiments (Adv. Mater. 2015, 27, 266; Electro. Acta 2017, 241, 433). This is the motivation that all the experimental data used for model training are done by ourselves with the same experimental and measurement conditions. To be clear, we modified statement to “Comparable training data used in SR analysis are of crucial importance for SR in order to produce accurate and useful mathematical formulas”.

9) in what sense the formulas mentioned in line 69 are "accurate"? Again, "accurate" needs a quantitative measure.

Response: In this work, “accurate” is quantitatively measured by the mean absolute errors (MAE) of descriptor-predicted V_{RHE} to experimental values, as shown in Fig. 3a. Here, we deleted ‘accurate’ in line 69 to avoid confusion at this place.

10) what is the precise meaning of the word "insightful" in the phrase "insightful formulas" (see line 69)?

Response: In the original version, “insightful” means that the formula can provide direct guidance for materials design. For example, μ/t is insightful since it helps design better perovskites, as discussed from line 122 to line 142 (original version). While, $(\frac{Q_A}{\chi_B} + 1.034 + \mu + (\frac{\mu}{t})^{0.5})^{0.5}$ as shown in Tab. 2, is not insightful, since it cannot easily provide guidance how to design better perovskites. To be precise, we modified "insightful formulas" to "useful formulas" in revision.

11) what is "structural flexibility" and what is "chemical flexibility" of perovskite structures stated in line 71? Do the authors refer to the fact that perovskites can take several different lattice structures (classified by symmetry groups) and that several different material compounds are called perovskites?

Response: Yes, referee’s understanding is exactly what we want to express here. "Structural flexibility/diversity" and "chemical flexibility" are often quoted as the important features of perovskites, for example, in the book of *Properties and Applications of Perovskite-Type Oxides*, L. G. Tejuca, J. G. Fierro, CSC Press (1992), the book of *Perovskites and Related Mixed Oxides: Concepts and Applications*, P. Granger, V. Parvulescu, S. Kaliaguine, W. Prellier, Wiley-VCH (2015) and the review of *Energy Environ. Sci.* 12, 442 (2019).

12) what are "obvious physical insights into the studied data sets" referred to in line 32?

Response: In the original version, we considered ML model as a ‘black-box model’ [Nat. Mach. Intel. 1, 206 (2019); *npj Comput. Mater.* 5, 83 (2019)]. The words that "ML does not provide obvious physical insights into the studied data sets" is equal to the statement that “ML is a black-box model”. In revision, we considered SR as an approach of interpretable ML method, deleted the words of "obvious physical insights into the studied data sets" and reorganized the introduction part.

13) what is "the most parsimonious way" in line 98?

Response: The parsimony in genetic programming is based on the assumptions that a relationship that requires the smallest number of genetic operation is most likely to be correct. Technically, it requires algorithm to achieve a descriptor in smallest number of genetic operation, reflected by the complexity in this work [see more details in references: Balancing Accuracy and Parsimony in Genetic Programming, B.-T. Zhang and H. Muhlenbein, *Evolutionary Computation* 3, 17 (1995) and the book of *Genetic Programming Theory and Practice II*, U.-M. O'Reilly, T. Yu, R. Riolo, B. Worzel, Springer (2005) (pp.283-299)]. Considering the word of 'parsimonious' is too technical and algorithmic, we deleted the words "the most parsimonious way" in revision to avoid possible confusion and focused our expression on scientific part.

The authors' statement in the first paragraph that "statistical ML does not provide obvious physical insights into the studied data sets" is dubious:

There is simply no need to apply any statistical ML or any other advanced method to provide "obvious" physical insights. If neglected obvious, this statement will be presumptuous in the sense that it disregards the variety of recent achievements by statistical ML in materials science.

Response: In the revision, we considered SR as a kind of interpretable ML method and this sentence has been revised to "However, the black-box model of ML is often criticized not able to provide new "physical laws", which limits its potential in certain cases.^{6,7}". The proper references [6,7] supporting this statement have been cited.

The authors state that "symbolic regression is a unique machine learning approach". Thus, how does the authors' machine learning approach differ from statistical ML, which the authors try to avoid?

The authors' statement that the symbolic regression "is different from statistical machine-learning approach, which bears a hidden black-box model and is difficult for physical interpretation" is not a reasonable argument and in particular, this is not a scientific argument publishable in Nature Communications!

Response: In the revision, we considered SR as a kind of interpretable ML method. We modified the statement throughout the manuscript (see highlighted version in resubmission) and hope referee would find that the revised statement is acceptable.

Notably, in ML fields, there are still discrepancies on the relationship between 'machine learning' and 'symbolic regression'. For example, Wang *et al.* consider SR as alternative method to ML [MRS Comm. 9, 793 (2019)], while Jin *et al.* consider SR as a kind of interpretable ML approach [arXiv:1910.08892v3 (2020)]. In this work, we did not intend to clarify those debates, but use the algorithms as tools to predict a simple descriptor and accelerate materials discovery. The scientific results and conclusions do not change in terms of different terminologies. We tried our best to make the terminologies and scientific description consistent and precise in the revision and we hope referee would find that the revised statement is acceptable.

The authors claim that "statistical ML is not easily accessed by many researchers, particularly by experimentalists". The authors further state that SR is "in contrast to statistical ML, which is a 'black-box'". However, the authors do not clarify that SR is more easily accessible than statistical ML. What do the authors mean exactly by the "black-box"? Given that, for instance, statistical ML using Kernel Ridge Regression is based on a supervised least-square fit with penalty term, statistical ML is not generally a "black-box". Therefore, the authors' statement in the manuscript's first paragraph is questionable.

Response: We thank referee for pointing out those confusions. In general, ML is often considered as a 'black-box model' [Nat. Mach. Intel. 1, 206 (2019); *npj Comput. Mater.* 5, 83 (2019)]. That is because most ML models, such as the KRR as referee mentioned, learn the parameters (like the weight, bias, regularization coefficient) of some hard-coded algorithms through data to predict the target value. For researchers who do not have any ML domain knowledge, what the model does is carrying out tasks based on input data and obtaining the target value. It is difficult for them to obtain any insights from the model, for example, guidance to design materials with high OER activity in this work. This is the meaning of 'black-box'.

Although SR is also one kind of supervised ML model, it aims to find mathematical formula of features from input data. This mathematical formula, referred here as descriptor, is easily understood and accessible by non-experts, as also supported by Referee #3.

We agree with the referee that all statistical ML models are not a "black-box". In revision, we soften our tone and reorganized the statement by considering SR as a kind of interpretable ML method. Notably, the scientific results and conclusions do not change in terms of different terminologies. We tried best to make the terminologies and scientific description consistent and precise in the revision.

The authors use the orbital filling, but without providing any definition of this quantity. How is the orbital filling mathematically defined in this study and how is it calculated in this study? Which orbitals are the authors referring to?

Response: In the original version, 'orbital filling' is the electronic occupancies of e_g orbital of transition metal B in perovskite ABO_3 , which was originally proposed by Suntvich *et al.* [Science 334, 1383 (2011)]. In the revision, we modified 'orbital filling' to ' e_g occupancy' to be consistent with the definition in original reference. In this study, we did not calculate ' e_g occupancy' but directly cite the values from original reference.

What are the physical (or geometrical) interpretations of the "octahedral" and "tolerance" factors, respectively? In the definition in line 94, the quantities r_A , r_B , r_O are changed to R_A , R_B , R_O in the Supporting Information without reasoning. To be precise, I suggest to provide the definition of the tolerance and octahedral factors as equations. The expression $t(\dots)$ is usually read as "t is a function of".

Response: To address the referee's concerns, we revised the expression of " $t(\dots)$ " and added one

sentence of “The tolerance factor t , defined as $\frac{r_A+r_O}{\sqrt{2}(r_B+r_O)}$ and octahedral factor μ , defined as r_B/r_O , are commonly-used features in machine-learning study of perovskites.” Meanwhile, we have written all the places of ionic radii to r_A , r_B , and r_C . We thank referee for pointing out this.

How are "weighted averaged ionic radii" in R_A and R_B defined? Providing all mathematical definitions is mandatory to make results reproducible.

Response: For example, for chemical formula $(A_xA_{1-x}^2)(B_yB_{1-y}^2)O_3$, the weighted averaged ionic radii of cation A is $r_A = x \cdot r_{A^1} + (1-x) \cdot r_{A^2}$. This definition has been supplemented in revised

Supporting Information. Original data for calculating the weighted averaged ionic radii and electronegativities for all the oxide perovskites are provided in Table S7 to ensure our results are reproducible.

In Fig. 2a, what is the precise meaning of "materials index"? What information and experiments include the "no. of experiments" in the plot in Fig. 2a? The quantity "no. of experiments" seems to not been mentioned anywhere in the main text.

Response: We thank the referee for pointing out this. The corresponding experimental information can be found in lines 76-81 (original version). To make it clear, we revised the caption of Fig. 2 as “a, the landscape of all V_{RHE} data produced by experiments, including eighteen conventional and five new perovskites (totally twenty-three perovskites listed as ‘Materials index’ with sequence shown in Tab. 1). Each perovskite has been made four samples and each sample has been measured three times (totally twelve measurements listed as ‘No. of experiments’).” to show the meaning of x- and y- axis of Fig. 2a.

In Fig. 3c and in line 46, what does e_g refer to?

Response: The d orbital of transition metal B in perovskite ABO_3 was split into t_{2g} and e_g orbitals under octahedral crystal field. In Fig. 3c, ‘ e_g electron’ means the electronic occupancy of e_g orbital, which was previously proposed as catalysis descriptor by Suntvich *et al.* [Science 334, 1383 (2011)].

Figures in the Supporting Information (and in the main text) should appear within the paragraphs, in which they are referenced.

Response: We have double checked all the figures in main text and supporting information and confirm that they are properly cited in the paragraph.

Symbolic regression (SR) is the central method in the theory part of the manuscript. However, SR is not described comprehensively and not presented appropriately: For details on SR, in the main text the authors refer to "Fig. 2b and Supplementary Information". In Fig. 2b, the authors again

refer to the Supplementary Information for details on SR. In the brief SR paragraph in the Supplementary Information, the authors in turn refer to the "Method part" in the main text. In the Method part in the main text, the authors in turn refer to Tab. S4 in the Supporting Information. At the end of the brief Method part about SR in the main text, the authors state that "the details of the symbolic regression approach are provided as Supplementary Information". Hence, this presentation of the SR method is confusing. Thus, I propose to include all relevant information about SR, which are needed to enable the reader to understand the SR comprehensively without additional sources, in the Method part in the main text.

Response: We thank the referee for careful reading and constructive suggestions. We significantly revised the Methods part and Supporting Information in revision. Now, the Methods part focuses on the setup of hyperparameters of *gplearn* code used in this work. Since the general information of SR is too long, we put it in Supporting Information. In this sense, the readers who just wanted to repeat our results by *gplearn* code can refer to the Method part and who wanted to understand how SR works can refer to the Supporting Information. We hope referee would find that such reorganization is easier to understand.

Regarding the SR method description and the associated flowchart in Fig. 2b:

- 1) What are the "432 sets of hyper-parameters"? How large are the sets? What are the hyperparameters? What do is the effect of the hyperparameters?

Response: In ML model, the model itself usually contains parameters like the weight, bias, regularization coefficient. Meanwhile, there are also parameters in machine learning algorithm to generate models. To avoid confusion of these two '*parameters*', the parameters to generate the ML model are called '*hyperparameters*'. The hyperparameters for SR in this work are now listed in Tab. 3. As shown in Tab. 3, we used grid search for 18 crossover probabilities from 0.5 to 0.95 with step of 0.025, 8 subtree mutation probabilities and 3 parsimony coefficients. Therefore, the grid number of hyperparameters combinations are $18 \times 8 \times 3 = 432$. We supplemented the explanation of each hyperparameter in Method part.

- 2) What are the "1000 selected formulas"? What do the formulas describe? Does "formula" (see also line 69) refer to mathematical or chemical formulas? How are the formulas generated or selected? This all remains absolutely unclear.

Response: We thank the referee for pointing out the confusion between 'chemical formula' and 'mathematical/analytical formula'. In revision, we have explicitly stated 'chemical formula' or 'mathematical formula' wherever the word of 'formula' is. The generation of mathematical formula and their evolution in genetic algorithm have been revised in Supporting Information. We hope the referee would find that the statement in revision is clear.

- 3) What are "random analytical formulas"? What means "random" here, what means "analytical" here, and (see point above) what means "formula" here?

Response: In the revision, we used 'mathematical formula' instead of 'analytical formula' to

accommodate referee's concern. The generation of mathematical formula and their evolution in genetic algorithm have been stated in revised Supporting Information. Basically, given the variables (Γ_A , Γ_B , N_d , χ_A , χ_B , Q_A , t , μ) and mathematical operators and functions (+, -, ×, ÷, $\sqrt{\quad}$), they can build tremendous mathematical formulas that cannot be exhausted. Therefore, genetic algorithm is able to generate a large number of random mathematical formulas (with random number of variables and operators, random combinations, random complexity).

- 4) What are "random analytical forms" in Fig. 2b? How are they "created"? Do the authors use "forms" as synonym for "formulas" and, if yes, why?

Response: Fig. 2b shows the flowchart of SR approach. Here, "random analytical forms" does not mean one particular mathematical formula but the general flowchart. We hope referee would find the revision of algorithm part is easier to understand. In genetic algorithm, mathematical formula has been represented by tree structures, facilitating the random generation and genetic operation. Referee is correct that we use "forms" as synonym for "formulas" in Fig. 2b for simplicity. To avoid confusion, we have change 'form' to 'formula' in Fig 2b.

- 5) How are "random tree structures" generated? Associated mathematical details of the method should be explained in the manuscript.

Response: "Random tree structures" are equivalent to "random mathematical formula" above, since the mathematical formula are in the form of tree structures in the genetic programming. As mentioned above, they are generated by well-established genetic algorithm. We used *gplearn*, a genetic programming code to generate tree structure (mathematical formula).

- 6) How is the "performance of formula [...] measured by mean absolute errors"? What is the precise meaning of "performance" in the context of "formula"?

Response: "Performance" here means "accuracy", which is measured by mean absolute errors (MAEs). MAE of descriptor (mathematical formula) for eighteen oxide perovskites is defined as $\sum_{i=1}^{18} \frac{|V_{i,RHE}^{pre} - V_{i,RHE}^{exp}|}{18}$, where $V_{i,RHE}^{pre}$ and $V_{i,RHE}^{exp}$ are descriptor-predicted and experimental overpotential for i -th perovskite. To be clear, we have modified this sentence to "The accuracy of formula is measured by mean absolute errors ...".

- 7) What is a "final solutions set"? What means "solution" here?

Response: As shown in Fig. 2b (simplified flowchart) and Fig. S11 (detailed flowchart), the best mathematical formula (with least MAE) in each generation has been put into the "final solution set". The terminology of "final solution" or "solution" can be referred to manual of *gplearn* code [<https://gplearn.readthedocs.io/en/latest/intro.html>]. Here, "final solution set" means the dataset of final solutions. To be clear, we changed "final solution set" to "Dataset of final solution" in revision.

8) How to "form 1000 new" "generic operations of crossover and mutation"? What means "generic" here?

Response: We are sorry that this is a typo. 'generic' should be 'genetic', which was corrected in revision.

9) What is the "fitness of each form" (see Fig. 2b)? How is "fitness" measured quantitatively and mathematically?

Response: 'fitness' or 'fitness metrics' is a terminology often used in genetic programming. In this work, MAE and fitness are the same. This was also claimed in line 244 (original version). To avoid confusion, we discarded the word of 'fitness' and only used 'MAE' in revision.

10) What is the "termination criteria" (see Fig. 2b)?

Response: In genetic programming, the code would produce mathematical formulas generation by generation (with each generation 5,000 mathematical formulas). If there is no "termination criteria", the code cannot stop. As in Tab. 3, we set "termination criteria" as 0.01 eV, which means that the code would stop when the MAE of best mathematical formula was less than 0.01 eV.

11) What are the "genetic operations" applied in Fig. 2b? Does Fig. 2b here refer to "crossover and mutation", which are mentioned in the Supporting Information?

Response: Yes. "genetic operations" in Fig. 2b means "crossover and mutation".

12) How are "descendants" added as stated in Fig. 2b? How are those "descendants" generated?

Response: The generation of mathematical formulas and their evolution in genetic algorithm have been supplemented in revised Supporting Information. As shown in Fig. S11, each generation includes 4000 randomly generated mathematical formulas and 1000 formulas inherited from the previous generation via genetic operation. The 1000 mathematical formulas are so-called "descendants" because they are derived from the previous generation by genetic operation.

13) In line 36, what are the "analytical forms that may have physical meanings"? What "physical meanings" do the authors refer to exactly?

Response: As mentioned above, we revised the statement of SR. In the revision, the sentence has been modified to "In contrast to 'black-box' model, SR delivers interpretable mathematical formulas that may provide direct guidance for materials design." to avoid confusion.

14) Regarding symbolic regression, why is it called "symbolic"? In what sense it is a "regression"?

Response: Here, symbol means the mathematical building blocks such as mathematical operators, analytical functions, variables and constants. Conventional regressions seek to optimize the parameters for a fixed model structure (for example fitting coefficients in a linear or polynomial model), while symbolic regression attempts to discover both model structures and model parameters. Therefore, the regression is not only on parameters or coefficients but also on symbols (+, -, ×, ÷, exp, sin, *etc.*). Notably, the approach of SR is not original contribution in this work, but an established method [John R. Koza; Martin A. Keane; James P. Rice, IEEE International Conference on Neural Networks. San Francisco: IEEE. pp. 191–198 (1993)]. In this work, we used SR as a tool, as implemented in *gplearn* code, to discover new catalysis descriptor.

15) What is the link between symbolic regression and the therein described genetic algorithm?

Response: Genetic algorithm is the algorithm to compose the building blocks as mentioned above to do regression of symbols and to search optimal mathematical formula. How genetic algorithm has been applied to SR was explained in revised Supporting Information. Basically, in genetic algorithm, mathematical formula has been represented by tree structures, facilitating the random generation and genetic operation. Genetic algorithm is one method of many. There are also other methods for the regression of symbols, for example recently-proposed Bayesian optimization [arXiv:1910.08892 (2020)].

16) Vague explanations in the manuscript such as "symbolic regression is a unique machine learning approach" are not helpful for understanding the method. What do the authors mean by "unique"?

Response: We have revised such kind of expression and hope referee would find the revised Method part and Supporting Information as well as the response above would help understand the approach of SR.

17) What are "best forms" stated in Fig. 2b? What does "best" refer to and how are "best forms" determined quantitatively? Is the activity of the perovskites determined in the theoretical approach and, if yes, how?

Response: Here, formula is not chemical formula but mathematical formula. We have revised 'form' to 'mathematical formula' in Fig 2b.

Since the SR method is not well described, it is not reproducible how the resulting perovskites are selected. Therefore, I suggest to present the theoretical method not in a general scheme, but in terms of how the perovskite structures are generated, evaluated, and optimized.

Response: Based on the response and changes above, we hope the referee would find the SR method is now well described in the revised version.

The authors state that the suggested perovskites are "among the oxide perovskite catalysts with the highest specific activities (Ref. 8)". This is imprecise: Which perovskites from their studies do the

authors compare to which perovskites in literature? What are the activities of both? Are all activities measured in the same experimental environment, and are the activities comparable and, if yes, why?

Response: This statement is based on the data shown in Fig. 3b and Fig. S7. In Fig. 3b, we compared V_{RHE} values of five new oxide perovskites (no. 19-23) to that of BSCF (no.17). The averaged V_{RHE} value for BSCF is 1.639 V and those for no. 19-23 are between 1.593 V and 1.673 V. Four of them (no. 20-23) have less V_{RHE} (high OER activity) than BSCF. This is the conclusion based on the comparison of our own data.

To test the comparability of our data, we compare our results with previous results [Science 334, 1383 (2011)] as shown in Fig S2. Both show the same trend of V_{RHE} for the same eight oxide perovskites, including BSCF.

In Fig. 3d, the octahedral/tolerance descriptor is applied on perovskites from literature. How does the descriptor describe the perovskites from this study? Why is this descriptor, which is even highlighted in the abstract, applied only on literature data, but without applying the descriptor on the data from this study?

Response: The application of octahedral/tolerance descriptor on the data of this study was shown in Fig. 3b. Moreover, the octahedral/tolerance descriptor was applied to the strategy of materials design (line 122-142, original version), which is crucial for this work to discover five new oxide perovskites.

In sum, regarding the above-discussed points I cannot recommend this manuscript for publication, in particular with regard to Nature Communications.

Response: We really appreciate referee's careful reading and valuable suggestions. Accordingly, we modified the language used throughout all the manuscript towards a more scientific description and reorganized discussions with all relevant information about SR in Methods and SI. A highlighted version has been resubmitted to help see the changes clearly. We hope that the referee would find the revised version acceptable for publications.

Reviewer #3 (Remarks to the Author):

I previously reviewed this manuscript for another NPG journal. The revision has greatly improved it and addressed my previous concerns well.

Weng et al. report an attractively simple structural descriptor for the oxygen evolution reaction of perovskite oxides, namely μ/t . The descriptor is identified among others possible choices by symbolic regression where simplicity is a criterion for selection. The authors use it to predict new perovskite catalysts that are synthesized, phase checked by XRD and then tested by cyclic voltammetry and chronoamperometry.

The main claims are that the work is based on the new descriptor μ/t and that the new catalysts are superior to BSCF. Both claims are well supported and the work is quite impressive due to the rational catalyst design and verification of the high activity of the newly predicted electrocatalysts. Moreover, the approach to identification of new descriptors is very promising. I believe the simple descriptor and the novel predictive approach are outstanding features of the work and they will be important for rational design of future perovskite catalysts in the electrocatalysis community as well as related catalysis communities where perovskites are used.

In this version, the statistical foundation of the identification of the new descriptor is easier to follow for non-experts.

The proposed descriptor μ/t is indeed new and very useful. It is limited to perovskite and perovskite-like structures, which are important materials classes for catalysis. The authors calculate μ/t for several current densities in their own systematic dataset and also for several previously reported systematic studies where the activity trends are correctly predicted. To test the generality of the new simple descriptor, I also asked my team to plot the OER activities of our literature database against μ/t obtained from Shannon's table where again the reported activities are anti-proportional to μ/t . Thus, I am confident that this descriptor will be used even though many others (such as eg occupancy) have been proposed before. It is superior to many other descriptors because tabulated values of the ion radii can be used which removes some ambiguity of obtaining the spin state and valence (eg occupancy) or calculating reaction free energies by DFT.

The authors rationalize the descriptor based on cation sizes and the perovskite stability. It is recommended to look for new catalysts at the boundary where the perovskite structure is predicted to become unstable (according to Goldschmidt's rules) but still adopts the perovskite experimentally. Monovalent A-sites are currently rarely used in perovskite oxide catalysts but the descriptor μ/t uniquely suggests their attractiveness and they may become more common due to the authors' work.

I was also very impressed that the authors not only predict a new descriptor but also systematically synthesize new perovskite oxides with new compositions and test their OER activity. The determined intrinsic activities of the new perovskite oxides rank among the best reported so far.

Response: We thank the referee for his/her valuable and encouraging praise. We are pleased to see our proposed descriptor is independently confirmed by experimental results in referee's group. It brought us more confidence that SR is a promising approach that may have real impact on materials design in future.

Minor points:

- 1) The information in the methods sections "OER characterization" and "electrochemical characterization" seems to be mainly duplicated. Please merge them.

Response: We have merged them into "OER characterization".

2) The eg descriptor is commonly attributed to the original research publication (Suntivich, Science 2011; ref. 7) and not later reviews of the Shao-Horn group. Ref. 14 could still be cited for an updated discussion.

Response: We have modified the citations accordingly.

3) Fig. 2a is very difficult to understand as well as crowded and thus unclear. Can it be plotted in an x-y plot, e.g. by using the average and standard deviation rather than experiment # as an axis? The data are also submitted as excel sheets where one can look up the individual values (which I find commendable).

Response: We thank the referee for the suggestion and Fig. 2a is accordingly revised to Fig. R6. It seems that Fig. 2b looks better by showing the feature of high-throughput experiments in this work.

Fig. R6 Data reorganization of Fig. 2a. There are eighteen conventional and five new perovskites (totally twenty-three perovskites listed as ‘Materials index’ with sequence shown in Tab. 1. The data are shown by using the average and standard deviation. For each perovskite, V_{RHE} values at five current densities of 50 $\mu\text{A}/\text{cm}^2$, 5 mA/cm^2 , 10 mA/cm^2 , 15 mA/cm^2 , and 20 mA/cm^2 are shown.

I support the publication of this manuscript in Nature Communications. The minor points could be addressed without further review.

REVIEWERS' COMMENTS:

Reviewer #1 (Remarks to the Author):

The authors have suitably addressed my criticisms, though I think it would be better if some of the discussion and analysis that was included in the response was also included in the manuscript or supporting information (even if the peer review file is made public). In particular, Figure R2, Figure R4, and Table R1 (and the associated discussion) seem valuable for readers should the same questions that occurred to me while reviewing also occur to them while reading.

-Chris Bartel